# Transcription of Clock Genes in Medulloblastoma

**DOI:** 10.3390/cancers17040575

**Published:** 2025-02-08

**Authors:** Jerry Vriend, Aleksandra Glogowska

**Affiliations:** Department of Human Anatomy and Cell Science, Rady Faculty of Health Sciences, Max Rady College of Medicine, University of Manitoba, Winnipeg, MB R3E 0J9, Canada; aleksandra.glogowska@umanitoba.ca

**Keywords:** brain tumor, clock genes, medulloblastoma, transcription, isochromosome 17, *USP2*, *CYR1*, survival-related genes

## Abstract

Since the expression of many genes varies throughout the day and night in various tissues of the body, we studied the expression of ‘clock’ genes in medulloblastoma (MB), a malignant brain tumor found primarily in children and young adults. Using publicly available data, we found that the core clock genes were expressed in MB tissue and that their expression was related to the four molecularly defined subgroups of MB: Group 3, Group 4, SHH, and WNT. The genetic aberration of isochromosome 17 in MB Groups 3 and 4 was associated with the over-expression of several core clock genes, including *CIPC* (Clock Interacting Pacemaker). The major biological pathways related to clock gene expression were the *ribosome* pathway and the *phototransduction* pathway. The two clock genes most related to patient survival were *CRY1* and *USP2*.

## 1. Introduction

Clock genes encode transcription factors and proteins that regulate behavioral and physiological rhythms that have daily patterns of approximately 24 h. They function as the biological timekeepers of circadian rhythms. The 2017 Nobel Prize in Physiology and Medicine was awarded to Hall, Rosbash, and Young for their work on clock genes and proteins and led to the transcription–translation feedback model as essential to account for the circadian oscillation of clock components [1]. Circadian rhythms in the transcription of clock genes occur in the suprachiasmatic nucleus (SCN) of the hypothalamus as well as in peripheral tissues [2,3]. Most peripheral tissues express clock genes [4]. The expression of clock genes in some peripheral tissues may be synchronized by the SCN [5], but may be independent of the SCN in others. Seven of the core clock genes have been reported to undergo a circadian oscillation in all the major tissues tested including the liver, kidney, lung, adipose tissue, muscle, heart, and brain [6]. The disruption of circadian rhythms can result in a variety of diseases. There are reports that the disruption of circadian rhythms may be involved in the development of several cancers including lung, breast, liver, and prostate cancer [7,8,9,10,11,12]. The disruption of the circadian clock is regarded as a risk factor for cancer [13,14], and circadian clock proteins have been proposed as therapeutic targets in cancer [9,15].

The core clock genes include genes for transcription factors *CLOCK*; *BMAL1* (also known as *ARNTL)*; *BMAL2* (also known as *ARNTL2*); *NPAS2*; the transcription repressor genes *PER1*, *PER2*, and *PER3* (*PERIOD 1*, *2*, and *3*); and the cryptochrome genes *CRY1* and *CRY2* [5,16]. The BMAL1/CLOCK complex stimulates the transcription of the period and cryptochrome genes by binding to E-Box elements in their promoters. The PER and CRY proteins in turn bind to CLOCK/BMAL1 forming a PER/CRY transcription/translation feedback loop (Figure 1) which drives the circadian clock [5].

In addition to the primary PER/CRY feedback loop, a secondary transcriptional/translational feedback loop regulating the core clock genes, the REV-ERB and ROR (Related Orphan Receptor) feedback loop, has been described and is regarded as part of the transcriptional structure of the circadian clock [5]. REV-ERB-alpha and REV-ERB-beta proteins are encoded for by the genes *NR1D1* and *NR1D2*. NR1D1 is a transcriptional repressor and nuclear heme receptor that has been reported to coordinate circadian rhythms of activity [17,18].

Finally, a third feedback loop was described by Cox and Takahashi [19] as a DBP/NFIL3 (D-Box binding transcription factor/Nuclear factor IL3 binding protein) loop. In the model of Cox and Takahashi, DBP and NFIL3 dimerized proteins bind to D-box elements of the *PERIOD*, *RORA*/*B*, and *NR1D1*/*2* genes to regulate transcription [19].

Several components of the ubiquitin–proteasome system have been found to regulate the expression of circadian clock genes [5], and are included in our analysis of MB clock gene expression. The best documented are the F-box ubiquitin ligase adaptors FBXL3 [20,21], FBXL21 [22,23,24], [25,26,27], FBXW7 [16,28], and the deubiquitinase USP2 [6,29,30,31,32,33]. *FBXL21* is highly expressed in the suprachiasmatic nucleus, the hypothalamic nucleus, which is the major pacemaker in mammals, whereas *FBXL3* and *USP2* are expressed in many tissues [22]. A mutation in *FBXL3*, referred to as *OVERTIME*, led to the discovery that the FBXL3 protein is required for the reactivation of *CLOCK* and *BMAL1* by facilitating the degradation of CRY1 and CRY2 proteins and resulting in the loss of negative feedback [34]. Another mutation of *FBXL3*, *AFTERHOURS*, slows the degradation of CRY protein [35]. The FBXL21 protein binds to the CRY proteins and antagonizes the action of FBXL3 on the degradation of these proteins [23].

Additional clock-related genes include *TIMELESS*, *CIPC* (Clock Interacting Pacemaker), and the clock regulator kinases *CSNK1δ* and *CSNK1ε*.

The current availability of public datasets on genome-wide gene transcription makes it possible to investigate the expression of circadian clock genes in various cancers and in their subgroups. Medulloblastoma (MB) datasets provide an interesting source in which to study the interaction of circadian rhythm genes and cancer since Group 3 MB tumors were initially delineated, in part, by the over-expression of photoreceptor and phototransduction genes [36,37,38,39,40].

In the current study, we investigate the expression of clock genes in public datasets of medulloblastoma. These include the Cavalli dataset [40] and the Swartling dataset, a reanalysis of 23 MB (including the Cavalli dataset), and non-tumor datasets, for batch variation [41]. The data enabled us to address the question of whether circadian clock genes were expressed in medulloblastoma tissue samples and if so, whether the transcription of these genes was differentially expressed among the four consensus subgroups of medulloblastoma, Group 3, Group 4, SHH, and WNT [36]. The Cavalli data allowed for further study of clock genes by the twelve subtypes within the four subgroups in this dataset; it also facilitated the examination of gene expression based on chromosome arm copy number variation. The Swartling data, moreover, enabled us to determine differences in the expression of clock genes between MB tumor tissue and non-tumor (NT) cerebellum.

The Cavalli dataset also had available survival data, which allowed us to determine whether the expression of any of the core clock genes in MB tissues was associated with patient survival. The data facilitated the identification of potential therapeutic targets among the clock genes that were associated with survival.

Reports relating clock genes and melatonin synthesis have been reported [42,43]. Therefore, we also examined the medulloblastoma datasets to determine whether the expression of the gene for the rate-limiting step in melatonin synthesis, *AANAT*, was related to MB subgroups.

Several clock genes, as well as *AANAT*, are located on chromosome 17, the chromosome associated with copy number gains of 17q in MB, and the chromosomal abnormality isochromosome 17q. The genes located on chromosome 17q include the clock gene *NR1D1*, the clock-related gene *CSNK1D*, and the gene encoding *AANAT*. The availability of copy number variation data in supplemental datasets of Cavalli et al. [40] facilitated the determination of whether the expression of clock genes in MB was related to the copy number gain of chromosome 17q.

Finally, based on the evidence that several clock proteins act as transcription factors or transcription regulators, we examined the most significant KEGG pathways associated with the genes whose transcription was correlated with that of each of the clock genes.

The hypothesis for this study is that clock genes are differentially expressed in the four major MB subgroups which may have an impact on clinical outcome and therapeutic potential. Our findings show major variations in clock genes are related to the isochromosome 17 aberration. Furthermore, our data show that variations in several clock genes are associated with survival.

## 2. Methods

### 2.1. Data Sources

Gene expression was mined from publicly available datasets of gene expression in MB, primarily the Cavalli dataset (GSE85217, *n* = 763) and the Swartling dataset (GSE124814, *n* = 1350 MB samples and 291 NT brain samples), a reanalysis of several datasets (including the Cavalli data). The medulloblastoma samples of the Cavalli dataset were collected with informed consent as a part of the Medulloblastoma Advanced Genomics International Consortium and approval from institutional review boards of the various institutions [40].

These datasets were made available through the R2 Genomics Analysis and Visualization Platform (R2 Genomics) (https://r2.amc.nl, last accessed 12 December 2024). Each of the datasets provided differential expression of genes for the four consensus subgroups of MB: the WNT, SHH, Group 3, and Group 4. The Cavalli dataset (*n* = 763) provided further information on gene expression in subtypes of each group, for a total of 12 subtypes, and also provided survival data. The Swartling dataset (*n* = 1641), a batch-corrected meta-analysis, in addition to providing gene expression of the four subgroups of MB, also included the gene expression data of non-tumor tissue derived from 291 normal cerebellum tissue samples. Subsequently, we refer to this group as the NT group. A detailed description of batch correction and dataset integration used in producing the Swartling dataset (in the R2 platform) is found in Weishaupt et al. [41].

Additional MB datasets found in the R2 Genomics Platform included the Pfister dataset, the Northcott MAGIC MB dataset, and the Gilbertson dataset. Additional NT datasets included the NT brain dataset of Berchtold and the NT cerebellum dataset of Roth.

### 2.2. Data Analysis

The R2 Genomics platform was also used for basic statistical analysis, including analysis of variance (anova), of differentially expressed clock genes and, in the Cavalli dataset, for the Kaplan–Meier analysis of survival data related to these genes. An anova was performed with and without a correction for multiple testing, the false discovery rate (FDR) correction.

In addition, a Cox proportional hazards regression analysis was performed through the R2 Genomics platform for the clock genes listed in Table 1. The Morpheus program (Broad Institute) was used to produce a heatmap and cluster analysis of clock gene expression in the Cavalli dataset (Figure 2).

Chromosomal copy number variation data, found in the supplemental tables of Cavalli et al. [40], facilitated the examination of clock gene expression as it relates to copy number gain of Chromosome 17q.

The KEGG pathways, associated with groups of genes correlated with each of the clock genes, were determined using the gene set analysis application in the R2 Genomics platform.

## 3. Results

### 3.1. Clock Gene Expression in Medulloblastoma Subgroups

Our list of clock genes included the core clock genes and genes that encode proteins that regulate clock gene expression. The differential expression of various clock genes between MB subgroups was highly significant by analysis of variance (anova). Table 1 shows the significance of MB group differences in the Cavalli dataset and the relative significance in the Swartling meta-analysis, which included an NT group in addition to the 4 MB subgroups.

Our gene expression results are discussed in the order of clock proteins presented in the 2021 review of Cox and Takahashi [19]: genes of the core clock proteins *CLOCK* and *BMAL1*, genes of the *PERIOD*/*CRYPTOCHROME* feedback loop (Figure 1), genes of the *REV-ERB* and *ROR* feedback loop, and genes of a third feedback loop described as a *DBP*/*NFIL3* loop. This is followed by the presentation of gene expression associated with the ubiquitin–proteasome system, which has been shown to have a key role in the circadian regulation of clock proteins; these genes include *FBXL3*, *FBXL21*, *FBXW7*, *BTRCP*, and *USP2*. Our focus is on the results with the greatest statistical significance, and on results that are confirmed across datasets. The differential expression of clock genes by the MB subgroup in the Cavalli dataset is illustrated in the heatmap and cluster analysis of Figure 2. The heatmap illustrates the differential expression of the clock genes by subgroup. The highest expression was found mainly in Groups 3 and 4, but depended on individual gene expression as shown in the figures of individual gene expression below.

Table 2 shows the statistics for the Kaplan–Meier survival curve scans (chi-squared and *p* values) up to 144 months, as well as the hazard ratios determined using the Cox proportional HR analysis. The most significant Kaplan–Meier scans (high vs. low gene expression) were found for the following clock-related genes: *USP2*, *CRY1*, and *CLOCK* (Table 2).

#### 3.1.1. CLOCK and BMAL Expression in MB Subgroups

The differential expression of ***CLOCK*** among the four major medulloblastoma (MB) subgroups was observed across multiple datasets. The analysis of the Cavalli dataset revealed significant differences in CLOCK expression (F = 77.07, *p* = 1.59 × 10^−43^), which were validated in the Pfister (F = 21.78, *p* = 2.21 × 10^−12^), Northcott (F = 52.76, *p* = 2.44 × 10^−20^), and recalculated Swartling datasets (F = 155.46, *p* = 6.27 × 10^−86^). The Swartling dataset, which included a normal tissue (NT) group, allowed for subgroup comparisons against NT (Figure 3). *CLOCK* expression was the lowest in Group 3 MB, with a highly significant reduction compared to NT (*p* = 1.25 × 10^−39^). In contrast, no significant difference in *CLOCK* expression was found between Group 4 and NT.

The CLOCK protein, together with its binding partner BMAL1, acts as a transcription factor regulating genes such as *PERIOD*, *CRYPTOCHROME*, and *NR1D1*/*NR1D2*. Low *CLOCK* expression, as observed in Group 3 MB, may disrupt cellular responses to DNA damage, as CLOCK has been implicated in this process. Notably, low *CLOCK* expression was associated with worse overall survival (Figure 3, Table 2). These findings suggest that the role of reduced CLOCK protein in the development and proliferation of Group 3 MB warrants further investigation.

The expression of ***ARNTL*** (coding for BMAL1) was also differentially expressed across MB subgroups in the Cavalli dataset (F = 59.29, *p* = 1.92 × 10^−34^) and validated in other datasets, including the Swartling dataset (F = 137.94, *p* = 1.88 × 10^−77^). While *ARNTL* expression was depressed in the SHH subgroup compared to NT (*p* = 2.8 × 10^−87^), it was significantly elevated in Group 3 MB (Figure 3). A strong statistical correlation between *ARNTL* and *RORA* expression (see below, Section 3.1.6) supports a potential functional relationship, such as *RORA* promoting *BMAL1* transcription.

The *CLOCK* homolog, ***NPAS2***, whose protein can dimerize with BMAL1 as an alternative to CLOCK, exhibited highly significant differential expression (F = 283.34, *p* = 1.59 × 10^−183^). Elevated NPAS2 expression was observed in Group 4 MB compared to other groups and NT, suggesting it may serve as a marker for Group 4 MB. NPAS2 has been reported as a tumor suppressor involved in DNA damage repair in breast cancer cells; however, its role in MB remains unexplored.

The expression of ***ARNTL2*** (coding for BMAL2) was lowest in the WNT subgroup (*p* = 1.82 × 10^−37^), with reduced *ARNTL2* expression associated with better survival outcomes (Chi = 12.91, *p* = 3.27 × 10^−4^) (Table 2). Interestingly, *ARNTL2* was specifically elevated in Group 4α, one of the twelve Cavalli subtypes.

#### 3.1.2. BHLHE40 and Survival in WNT MBs

***BHLHE40* (also known as *DEC1*)**, a transcriptional repressor that inhibits the transcriptional activity of the CLOCK/BMAL complex [44], was differentially expressed at a highly significant level (F = 209.02, *p* = 7.20 × 10^−144^). The expression levels of *BHLHE40* were elevated in the WNT subgroup compared to other subgroups and normal cerebellar tissue (NT) (Figure 3).

Our analysis revealed that this upregulation of *BHLHE40* in the WNT subgroup aligns with observations in other cancers, as reported by Kiss et al., who listed various malignancies with increased BHLHE40 expression [45]. This finding adds the WNT subtype of medulloblastoma (MB) to the list of cancers where *BHLHE40* is upregulated.

The Kaplan–Meier survival analysis (Table 2) demonstrated that elevated *BHLHE40* expression is associated with improved survival outcomes in WNT MB patients, further highlighting its potential role as a prognostic marker.

#### 3.1.3. Cryptochrome (CRY) and Period (PER) Gene Expression

Cryptochrome proteins are required to maintain circadian rhythmicity in mammals. Cryptochrome proteins are blue light photoreceptors in plants and in insects including Drosophila, but knockout experiments in mice lead to the conclusion that they do not function as direct photoreceptors in mammals [46]. The differential expression of ***CRY1*** (F = 160.65, *p* = 1.29 × 10^−80^) and ***CRY2*** (F = 35.25; 2.48 × 10^−21^) was noted among the four major subgroups of MB in the Cavalli dataset and in the Swartling dataset. Compared to the NT group in the Swartling dataset, *CRY1* expression (Figure 4A) was elevated in Group 3 MB by *t*-test (*p* = 5.73 × 10^−48^), but depressed in the SHH (*p* = 4.81 × 10^−47^) and WNT (*p* = 7.49 × 10^−17^) groups. *CRY2* (Figure 4B) expression was depressed in Group 3 (*p* = 5.76 × 10^−18^), SHH (*p* = 1.41 × 10^−46^), and WNT (*p* = 3.06 × 10^−30^) groups compared to the NT group.

Among the 4 MB subgroups, the highest expression of *CRY1* was found in the Group 3 MBs (Figure 4A) (*p* = 2.77 × 10^−56^). Individuals with high levels of *CRY1* were mostly less than 10 years of age (Figure 4C). High expression was associated with poor survival (Table 2).

PERIOD proteins adjust circadian rhythms to changes in photoperiod [47]. They function as transcriptional repressors in the transcription/translation feedback loop (Figure 1). The differential expression of the period genes, ***PER1*** and ***PER2***, was observed between the MB subgroups in the Cavalli dataset and in the Swartling dataset (Figure 5). Compared to the NT group in the Swartling dataset, the major statistical differences were a depression of *PER1* expression in the WNT subgroup (*p* = 2.34 × 10^−20^) and in Group 4 (*p* = 5.00 × 10^−20^). The decreased expression of *PER2* expression was observed in the WNT group (*p* = 1.72 × 10^−32^) and in Group 3 (*p* =5.29 × 10^−31^) groups compared to the NT group.

#### 3.1.4. TIMELESS and MB

The ***TIMELESS*** gene codes for a protein, TIM, that inhibits the activation of the PER1 gene and regulates minor changes in circadian rhythms by interacting with PER2 and CRY2 proteins [48,49]. It also contributes to the DNA damage response [50]. In the Swartling dataset, *TIMELESS* expression was substantially elevated in the MB groups G3, G4, and WNT compared to the non-tumor group (Figure 6). In the Cavalli dataset, high expression was associated with worse survival based on the Kaplan–Meier analysis (Table 2).

#### 3.1.5. CSNK1D and Chromosome 17q

***CSNK1D***, a gene located on chromosome 17q, encodes a casein kinase which phosphorylates PER1 and PER2 proteins. Phosphorylation, together with ubiquitination, enables the degradation of these proteins by the proteasome [51] (Figure 1). Figure 7 shows an increase in the expression of *CSNK1D* in Groups 3 and 4 and a major decrease in the expression of *CSNK1D* in the SHH subgroup compared to the other subgroups, including the NT group. Our analysis found that the high expression of this gene was statistically associated with copy number gain of chromosome 17q; its expression was elevated in individuals with 17q copy number gain (Table 3).

#### 3.1.6. REV-ERB and RORA Expression

The **ROR** (*RORA*, *RORB*, and *RORC*) and NR1D (*NR1D1* and *NR1D2*) genes encode proteins that bind to ROR-binding elements on the *BMAL1* (ARNTL) gene. RORA and REV-ERBα compete for these binding sites, regulating *BMAL1* transcription [19,52].

The most notable finding was the significant depression of *RORA* expression (Figure 8A) in the SHH group compared to the normal tissue (NT) group (t = 17.53, *p* = 3.00 × 10^−^⁵^7^). *RORA* expression was strongly correlated with *BMAL1* (*ARNTL*) expression in the Cavalli dataset. Among 18,267 gene combinations analyzed, RORA had the most significant correlation with *BMAL1* expression (r = 0.70, *p* = 3.19 × 10^−111^), supporting the role of *RORA* in promoting *BMAL1* transcription (Figure 8B) [53].

*RORA* also functions as a component of the second major feedback pathway regulating clock genes (Figure 8C). Its association with the *phototransduction* pathway was particularly striking. A correlation analysis revealed that the KEGG pathway most statistically associated with *RORA* expression was the *phototransduction* pathway, including the genes encoding proteins of the phototransduction cascade [54] (Table 4).

Seven genes of this pathway—*CNGB1*, *GNAT1*, *GRK1*, *GUCY2D*, *PDE6B*, *PDE6G*, and *RCVRN*—were highly correlated with *RORA* (r > 0.50) (Table 4). These genes were markedly over-expressed in the MB Group 3 alpha subtype in the Cavalli dataset (Appendix A). Furthermore, *AANAT*, the gene encoding the rate-limiting enzyme for melatonin synthesis, was also over-expressed in Group 3 MBs (see below, Section 3.3). The KEGG pathway statistically associated with *AANAT* and *USP2* expression was represented by the same four phototransduction-related genes (*GNAT1*, *GRK1*, *GUCY2D*, and *RCVRN*).

These findings are consistent with prior research by Hooper et al. [55], who compared Group 3 MB expression profiles to normal molecular developmental events. They found that Group 3 MB expression profiles resembled those of rod cell precursor cells at around 15 weeks of human retinal development. Hooper et al. described a neoplastic switch from granule cell precursors to rod cell precursors in a Group 3 MB cohort, suggesting a molecular relationship between Group 3 MBs and pineal tumors.

The refined classification of Group 3 MBs into subtypes—Group 3 alpha, beta, and gamma—in the Cavalli dataset highlights the increased expression of phototransduction genes in the Group 3 alpha subtype (Appendix A). This suggests that Group 3 alpha MBs may originate from photoreceptor precursors. Further investigation is needed to test this hypothesis and better understand the developmental origins of Group 3 alpha MBs.

#### 3.1.7. THRA/NR1D1, NR1D2, and Their Implications in Medulloblastoma (MB)

***NR1D1*** and ***NR1D2*** encode the proteins REV-ERBα and REV-ERBβ, respectively, which play key roles in circadian rhythm regulation [56]. The differential expression of the *THRA*/*NR1D1* locus (F = 423.43, *p* = 1.43 × 10^−181^) was observed across MB subgroups, with the highest expression in Group 4 MB (Figure 9). This locus includes two genes, *THRA* (Thyroid Hormone Receptor A) and *NR1D1*, which are transcribed from the opposite strands of DNA on chromosome 17q21.1 [57]. This arrangement complicates the interpretation of gene expression data, as datasets like Cavalli aggregate their expression into a single locus labeled *THRA*/*NR1D1*. Meanwhile, the Swartling dataset lists only *THRA*, further limiting the clarity of *NR1D1*-specific data.

The high expression of *THRA*/*NR1D1* in Group 4 MB aligns with the frequent occurrence of isochromosome 17q in this subgroup, particularly in the Group 4 beta subtype, where over 95% of the cases exhibit this chromosomal abnormality [40]. The statistical analyses revealed that *THRA*/*NR1D1* expression in Group 4 MB is strongly associated with copy number gain of chromosome 17q (Table 3). The Kaplan–Meier analysis showed a significant correlation between high *THRA*/*NR1D1* expression and poor survival (*p* = 1.84 × 10^−3^) (Table 2), suggesting its potential as a prognostic marker.

*NR1D1* (REV-ERBα) is a transcriptional regulator that inhibits *BMAL1* transcription [19,58] and regulates numerous circadian-expressed genes across multiple chromosomes (Figure 8C). The CLOCK/BMAL1 protein complex activates *NR1D1* transcription [59], further linking this gene to circadian rhythm pathways. High levels of NR1D1 protein (REV-ERBα) in Group 4 MB could lead to the suppression of BMAL1 activity, disrupting downstream transcriptional networks.

For *NR1D2*, the most significant difference by subgroup was observed as a depression in expression in the SHH subgroup compared to normal tissue (*p* = 3.86 × 10^−29^) (Figure 9).

The high level of differential expression of the *THRA*/*NR1D1* reporter (Hugene 11t platform probe 8007008) in Group 4 MB suggests it could serve as a biomarker for this subgroup (Table 1, Figure 9). This elevated expression likely reflects the impact of isochromosome 17q on transcription levels.

The downregulation of REV-ERBα (*NR1D1*) has been observed in various cancers, as reviewed by Gomatou et al. [60]. However, there is limited literature specifically linking *NR1D1* or its encoded protein REV-ERBα to MB. Our findings that high *THRA*/*NR1D1* expression is associated with poor survival in Group 4 MB underscore the need for follow-up studies. Exploring the role of REV-ERBα as a therapeutic target may yield new insights into treatment strategies for Group 4 MB.

#### 3.1.8. DBP (D-Box of Albumin Promoter) and NFIL3 Gene Expression

The differential expression of ***DBP*** (D-box binding protein) (F = 55.93, *p* = 1.14 × 10^−32^) was highly significant in the Swartling dataset, with the lowest expression values observed in the SHH group (Figure 10A). Compared to the NT group in the Swartling dataset, the depression of *DBP* in the SHH group was particularly pronounced (t = 15.65, *p* = 1.54 × 10^−47^).

***DBP*** gene transcription is a known target of the CLOCK/BMAL1 dimer complex. Both DBP and NFIL3 proteins bind to the D-box elements of the promoters of the selected clock genes, playing opposing roles in their regulation. Studies by Ueda et al. [61] and Keniry et al. [62], as reviewed by Cox and Takahashi [19], highlight these opposing effects on the D-box elements of clock gene promoters (Figure 10B).

*DBP* and *NFIL3* encode proteins that are components of the third clock gene feedback pathway (Figure 10) described by Cox and Takahashi [19]. The correlation analysis of *DBP* (Table 4) reveals that the most significant biological pathway associated with the *DBP* gene correlates is the ***GABAergic***
**pathway**, which has been linked to Group 3 MB in the 2012 consensus report [36]. Elevated *DBP* expression is illustrated in Group 3 MB (Figure 10A), suggesting that DBP may contribute to defining this subgroup.

While NFIL3 expression was not available in the Swartling dataset, in the Cavalli dataset, ***NFIL3*** expression was also the lowest in the SHH group compared to the other three subgroups (F = 106.98, *p* = 9.10 × 10^−58^) (Figure 10C). The highest transcription of *NFIL3* was observed in Groups 3 and 4 MB in the Cavalli dataset (Figure 10C). High expression of *NFIL3* was associated with poor survival (Table 2). Zeng et al. have reviewed the abnormal expression of *NFIL3* in various cancers and proposed NFIL3 as a therapeutic target [63]. However, to date, neither the *NFIL3* gene nor its protein has been specifically studied as a therapeutic target in MB.

#### 3.1.9. Clock Interacting Pacemaker (CIPC) (Also Known as KIAA1737)

*CIPC* expression is included in the present study due to the report that the CIPC protein binds directly to the CLOCK/BMAL1 complex [64] and due to its extremely high statistical differential expression by MB subgroups (Swartling, F = 379.47, *p* = 1.67 × 10^−150^; Cavalli, F = 587.87, *p* = 2.87 × 10^−240^). Compared to the NT group in the Swartling dataset, *CIPC* was elevated in Group 4 (*p* = 1.29 × 10^−48^), but depressed in the SHH (*p* = 1.28 × 10^−83^) and WNT (*p* = 1.46 × 10^−46^) groups (Figure 11A). Among the clock genes, a high correlation of *CIPC* and *FBXL21* was noted (r = 0.79, *p* = 7.75 × 10^−164^). Figure 11B shows the correlation of *CIPC* and *FBXL21*, with subgroup expression identified by color. High expression of both genes was associated with copy number gain of chromosome 17q (Table 3).

### 3.2. Ubiquitin–Proteasome Pathway Regulation of Clock Genes in MB


**FBXL3**


The differential expression of *FBXL3* was significant in the Swartling dataset (F = 55.21, *p* = 1.31 × 10^−43^ (Figure 12A). By *t*-test, the most significantly different expression of *FBXL3* from the normal group was depression in the WNT group.


**FBXL15**


Figure 12B shows that the expression of *FBXL15* was depressed in the SHH MB subgroup compared to the other three groups as well as to the NT group.


**FBXL21**


The expression values of *FBXL21* were not available in the Swartling dataset (due to its classification as a pseudogene in humans), preventing a comparison of expression values to the NT group. However, using the Megasample application of the R2 Genomics program, we were able to compare the expression of *FBXL21* in medulloblastoma (Pfister and Gilbertson datasets) to normal brain (Berchtold) and normal cerebellum (Roth dataset) using the same gene chip (Affytmetrix u133p2) and the same reporter (1555412_at). This analysis showed that *FBXL21* expression was elevated in MB compared to that of the NT group (Figure 13A).

The differential expression of *FBXL21* transcription was also highly significant in the Cavalli dataset (F = 289.07, *p* = 4.14 × 10^−125^), with minimal expression in the SHH MB group (Figure 13B); the expression of *FBXL21* in Group 3 was 25-fold greater than in the SHH group, while expression in Group 4 was approximately 35-fold greater than in the SHH group. This over-expression is primarily in children less than 15 years of age (Figure 13C). The findings of elevated expression of *FBXL21* in Groups 3 and 4 were confirmed in the Northcott MAGIC MB dataset, in the Pfister MB dataset, and in the Gilbertson dataset, all available in the R2 Genomics platform.


**FBXW7**


*FBXW7* expression was depressed in all four subgroups compared to the NT group in the Swartling dataset (Figure 14A). However, *FBXW7* expression was specifically elevated in one of the SHH MB subtypes in the Cavalli dataset, the SHH gamma subtype (Figure 14B), a subtype whose subjects are infants of less than 3 years of age. The SHH gamma subtype was the only subtype in which the expression of *FBXW7* was significantly elevated compared to all the other subtype groups, including NT controls. The Kaplan–Meier analysis showed that the high expression of *FBXW7* was protective (Table 2).


**MYCBP2**


*MYCBP2* is a gene that encodes a ubiquitin ligase (also known as PAM, a protein associated with *MYC*) reportedly necessary for normal CNS development [65] and required for the degradation of the *NR1D1* encoded protein REV-ERBα [66]. The expression of *MYCBP2* was highest in Cavalli Group 4 and lowest in the WNT group. The differential expression of this gene was among the highest statistical significance. (Figure 15; Table 1).


***USP2*—a gene encoding a deubiquitinase for clock genes**


**USP2** is an enzyme that deubiquitinates various proteins, including BMAL1, CRY1, and PER1 [29,30,33], and regulates the distribution of PER1 between the nucleus and cytoplasm [33]. The elevated expression of the *USP2* gene was associated with Group 3 MB (Figure 16). High expression of *USP2* was associated with poor survival (Chi-squared = 21.73, *p* = 3.14 × 10^−6^). By Cox proportional hazard ratio (HR) analysis of all the clock genes in Table 2, USP2 had the most significant HR (*p* = 3.5 × 10^−5^).

### 3.3. AANAT Is Over-Expressed in MB Group 3

Since melatonin has been associated with circadian rhythms and clock gene expression, we examined the expression of the gene encoding the rate-limiting enzyme of melatonin synthesis, *AANAT* (Aralkylamine N-Acetyltransferase) [67] in the MB subgroups. While *AANAT* data were not available in the Swartling dataset, the Cavalli data show that *AANAT* expression was significantly greater in Group 3 MBs than in the other groups (Figure 17A). More specifically the elevation in *AANAT* expression was observed primarily in a subtype of Group 3 MBs, the Group 3 alpha subtype. *AANAT* expression was elevated primarily in young children (Figure 17B).

### 3.4. Clock Genes and Survival in the Cavalli Dataset

Several clock genes were significantly associated with survival in the Cavalli dataset. Table 2 shows the Chi-squared and *p* values for the Kaplan–Meier analysis of survival curves (high vs. low scan) of clock-related genes in order of significance. Also included in Table 2 are the significant hazard ratios, as determined by the Cox proportional hazard analysis.

The three most significant Kaplan–Meier scans were of *CRY1*, *USP2*, and *CLOCK* expression. The Kaplan–Meier analysis showed that high expression of *CRY1* and of *USP2* was associated with poor survival (*CRY1*, Chi-squared = 22.33, *p* = 2.29 × 10^−6^ *USP2*, Chi-squared = 21.73, *p* = 3.14 × 10^−6^) (Table 2, Figure 18); high expression of *CLOCK*, on the other hand, was associated with better survival (Chi-squared = 20.21, *p* = 6.96 × 10^−6^). The expression of *CRY1*, *USP2*, and *CLOCK* were also associated with significant hazard ratios (Table 2).

The expression of *ARNTL2* (*BMAL2*) and *CSNK1D*, was also associated with significant Kaplan–Meier scans as well as with significant hazard ratios (Table 2). High expression of both of these genes was associated with worse survival (and low expression with survival protection).

The expression of *TIMELESS*, *FBXL21*, and *BTRC* showed significantly different (high vs. low expression) Kaplan–Meier scans, but no significant hazard ratios. At a lower level of significance, high expression of *RORB* and *NFIL3* were associated with worse survival (Table 2). High expression of *CIPC* was also associated with lower survival, albeit at a low level of significance (*p* = 0.046).

### 3.5. Clock Genes Correlates: Pathway Analysis in the Cavalli Dataset

Table 4 shows the correlates for each of the clock genes and the most significant KEGG pathways associated with them. The number of correlates of clock genes (at r > 0.50) varied from zero to over 1000. The highest number of significant correlates were for the clock-related gene loci *CIPC* (1063 genes), *THRA*/*NR1D1* (1041 genes), *FBXL21* (605 genes), and *MYCBP2* (183 genes).

The KEGG analysis identified the most over-represented pathway for the correlates of each of these four genes as the *ribosome* pathway (*CIPC*, *p* = 1.34 × 10^−73^; *THRA*/*NR1D1*, *p* = 1.44 × 10^−80^; *FBXL21*, *p* = 2.32 × 10^−74^; *MYCBP2*, *p* = 1.34 × 10^−48^). The expression of these four genes was negatively correlated (r > 0.50) with numerous genes for ribosome subunits (*CIPC*, 57 genes; *THRA*/*NR1D1*, 59 genes; *FBXL21*, 44 genes; *MYCBP2*, 29 genes). It should be noted that the most over-represented KEGG pathway for all the survival-related genes (*p* < 0.001 by Kaplan–Meier analysis) in the Cavalli dataset was the *ribosome biogenesis* pathway.

The KEGG pathway of *phototransduction* was the most significant pathway associated with the correlates of *RORA* (with seven contributing genes), correlates of *USP2* (four contributing genes), and for *AANAT* (four contributing genes). For each of these three genes, the correlates included the phototransduction genes *GNAT1*, *GRK1*, *GUCY2D*, and *RCVRN*, genes whose expression was selectively elevated in the Cavalli MB subtype, Group 3 alpha. Numerous phototransduction genes were over-expressed in the Group 3 alpha subtype in the Cavalli dataset; they include the rhodopsin gene (*RHO*) and the gene encoding phosducin (*PDC*). A detailed analysis of the enhanced expression of the genes coding for proteins of the phototransduction cascade in MB is beyond the scope of the present manuscript.

In addition for each of the three genes, *RORA*, *USP2*, and *AANAT*, the second most significant pathway associated with their correlates was the *GABAergic synapse*, including the four GABAergic-related genes, *CACNA1F*, *GNB3*, *GNB5*, and *TRAK2*.

### 3.6. Clock Gene Expression Is Related to Copy Number Gain of Chromosome 17q

A major statistical factor determining the expression of clock genes was the copy number gain of chromosome 17q (Table 3). This would be expected for genes located on chromosome 17q, such as the *THRA*/*NR1D1* locus, which showed the most significant effect of 17q copy number gain (*p* = 1.57 × 10^−84^) vs. normal 17q copy number. The increase in expression was substantial (Table 3). The data, however, show that increased expression associated with the copy number gain of chromosome 17q is also found for clock genes not on chromosome 17q, including *CIPC* (chr 14q), *FBXL21* (chr 5q), and *MYCBP2* (chr 13). Table 3 shows the statistical effect of 17q copy number gain on clock gene expression in order of significance by *t*-test.

*PER1* is located on chromosome 17p (17p13.1), on the short arm. In isochromosome 17, a copy of chromosome 17p is lost concurrently with a gain of 17q [68]. This would account for the reduction in *PER1* expression in individuals with a copy number gain of 17q in Table 3.

## 4. Discussion

It has been estimated that the circadian clock regulates up to 50% of the transcription in eukaryotes [69]. Evidence for the role of circadian clock genes in cancer is gradually emerging [70,71]. Here, we present the gene expression profiling of clock genes in the four major subgroups of medulloblastoma; furthermore, the availability of the batch-adjusted meta-analysis dataset of Weishaupt et al. [41] enabled a comparison of gene expression to that of NT brain tissue. We show that the prognostic significance of the selected clock genes depends on the molecular subgroups of MB defined (Group3, Group 4, SHH, and WNT) and to some extent, on the 12 molecular subtypes reported by Cavalli et al. [40]. While we observed significant MB subgroup differences in expression for most the of clock genes, we focus the discussion on those that are associated with the transcription of other genes, those that are most significantly associated with survival, and those that have potential therapeutic relevance.

The transcription–translation feedback model provides a functional paradigm that explains circadian variations in clock genes [1]. A number of the clock genes are active as transcription factors or transcription repressors. Querying Cytoscape for the molecular functions of these genes, we found the most significant GO (Gene ontology) molecular term associated with the genes of Table 1 was *E-box binding*. The reviews of Cox and Takahashi [5,19] illustrate the role of E-box binding in clock-controlled genes. The E-box regulators BMAL1, CLOCK, CRY1, and PER1, as proteins (Figure 1), play a role in several cancers [72,73,74], and MB should be added to the list of those cancers. E-box binding transcription factors (EBTFs) in cancer have been reviewed by Pan, Watt, and Kay [75].

Cox and Takahashi point out that the characterization of the human clock mechanisms is not nearly complete [19]. Our analysis of clock gene expression, showing MB group-specific variations in clock genes, suggests that transcription–translation feedback loops in clock genes are present in MB tissue and that they operate differentially in the tissues of the 4 consensus MB subtypes. We also identify the clock genes whose expression is related to copy number gain of chromosome 17q, a common genetic aberration in Groups 3 and 4 MB. Furthermore, our correlation analysis shows the major biological pathways associated with the expression of clock genes The present results suggest that the MB subgroups provide novel in vitro models for further study of the regulation of clock genes and proteins, and furthermore show that the expression of some of the clock genes are strongly associated with patient survival in MB.


***CRY1* and *USP2* and survival in Group 3 MB**


The gene expression data (Figure 4A) highlight *CRY1* as a defining factor in Group 3 medulloblastoma (MB), particularly in pediatric cases (Figure 4C). High *CRY1* expression, noted in Group 3—the MB group with the worst survival—correlates with poor prognosis (Table 2). *CRY1*, a circadian rhythm regulator, is also implicated in DNA repair and tumorigenesis in cancers such as prostate cancer, making it a potential therapeutic target in Group 3 MB. High CRY1 levels likely disrupt BMAL1/CLOCK-induced transcription, further contributing to the malignancy.

CRY1 stability is regulated by ubiquitination and deubiquitination (Figure 1). USP2, a deubiquitinase, stabilizes CRY1 by protecting it from proteasomal degradation, enhancing its negative feedback role in the circadian loop. *USP2* is over-expressed in Group 3 MB, particularly in young children, and is linked to survival outcomes (Table 2). The pathway analysis (Table 4) indicates that *USP2* correlates significantly with phototransduction and visual perception pathways, suggesting its circadian function is dysregulated in MB.

As an oncogene, *USP2* has been suggested as a therapeutic target in cancers. USP2 inhibitors have been reported [76,77] and could be tested for their effects on MB tumorigenesis and tumor cell proliferation. Additionally, the SCF-FBXL3 ubiquitin ligase complex plays a role in CRY1 degradation, with mutations in FBXL3 leading to CRY1 accumulation. Targeting the CRY1-USP2 feedback loop (Figure 1) offers a promising avenue for therapeutic intervention in Group 3 MB. These findings can be validated through gene expression analysis using qRT-PCR and RNA-seq, followed by assessing protein levels using standard Western blot technique or more advanced proteomics assays. Additionally, functional studies could be conducted using medulloblastoma cell lines to further investigate the impact of these findings. The observed upregulation of USP2 may contribute to increased CRY1 stabilization, leading to enhanced DNA damage repair and the inhibition of apoptosis. Evaluating the effects of these alterations on cell proliferation and migration will provide valuable insights into their role in tumor progression. Furthermore, drug sensitivity assays will be crucial in exploring potential therapeutic approaches by testing USP2 inhibitors and assessing their effects on cell viability, apoptosis (via Annexin V/PI staining), and tumorigenic properties.


**PERIOD genes and CSNK1D in MB: role of phosphorylation**


The *PER1* and *PER2* genes encode transcriptional repressors that inhibit CLOCK/BMAL (Figure 1). The USP2 protein binds and deubiquitinates PER1, regulating its nuclear localization [31,78]. The ubiquitination of period genes is mediated by the BTRC ligase [19]. *CSNK1D*, which encodes CK1δ, phosphorylates PER1 and PER2 proteins prior to their degradation [51]. The over-expression of *CSNK1D* is observed in various cancers [79], and in Group 3 and 4 MB (Figure 7), with high expression linked to poor survival (Table 2). The inhibition of CK1δ using PF-670462 has shown potential in reducing *PER1*, *PER2*, and other clock genes in rats [80] and is suggested as a cancer therapeutic agent [79].


**The TIMELESS connection to MB**


The *TIMELESS* gene, encoding the TIM protein, interacts with PER2 and CRY2 [48,49]. Elevated *TIMELESS* expression is found in Group 3, Group 4, and WNT MB (Figure 6), and high expression correlates with poor survival (Table 2). The over-expression of *TIMELESS* is linked to poor prognosis in various cancers [49] and has been suggested as a therapeutic target. TIMELESS also contributes to DNA stability during replication [50], with the pathway analysis (Table 4) highlighting the Fanconi anemia pathway, associated with medulloblastoma risk [81].


**Clock Genes, Ribosome Connection, and Copy Number Gain of Chromosome 17q**


Our analysis (Table 4) shows that the expression of four clock gene loci—*THRA*/*NR1D1*, *CIPC*, *FBXL21*, and *MYCBP2*—highly correlates with the ribosome biogenesis pathway. High expression of these genes inversely correlates with ribosome subunit gene transcription, with *THRA*/*NR1D1* showing the strongest negative correlation (59 ribosome subunit genes). These four genes are most highly expressed in Group 4 MB (Table 1), suggesting that the dysregulation of clock genes affects ribosome biogenesis.

*THRA*/*NR1D1*, located on 17q, is elevated in individuals with 17q copy number gain (Table 3). Similar changes were noted for the expression of *CIPC*, *FBXL21*, and *MYCBP2*. We propose the hypothesis that the isochromosome 17q aberration in MB leads to elevated clock gene expression and reduced ribosomal protein gene transcription.

*NR1D1* (REV-ERBα) and *CIPC* influence CLOCK/BMAL1-induced transcription (Figure 1 and Figure 8), with CIPC inhibiting this pathway [64]. Jouffe et al. [82] and Pelletier et al. [83] highlighted the coordination between clock genes and ribosomal biogenesis. Our findings suggest that clock genes are linked to subgroup-specific ribosome expression in MB, supporting the view that MB cell lines could provide insights into ribosome biogenesis. Our Kaplan–Meier survival analysis indicates that the ribosome biogenesis pathway is the most significant KEGG pathway associated with all the survival-related genes in the Cavalli dataset.


**Clock genes, the casein kinase connection, and copy number gain of chromosome 17q**


*CSNK1D1* is one of two clock gene loci located on chromosome 17q. The expression of both *THRA*/*NR1D1* and *CSNK1D1* is associated, at a very high statistical level, with a 17q copy number gain (Table 3). *CSNK1D1* encodes a kinase protein that can phosphorylate a number of proteins, including the PERIOD proteins, and as such, has been described as a major non-transcriptional regulator of circadian rhythms [84]. We suggest further study of CK1δ inhibitors (particularly those that cross the blood–brain barrier) as potential therapeutic agents in individuals with copy number gain of chromosome 17q.


**AANAT another chromosome 17q gene associated with phototransduction**


Since melatonin has been associated with circadian rhythms and reported to interact with BMAL1 [67], we examined the expression of *AANAT*, the gene encoding the rate-limiting enzyme in the synthesis of melatonin. *AANAT* is another gene located on chromosome 17q. However, no significant increase in expression was found in individuals with a copy number gain of 17q (Table 3).

The expression of *AANAT* was elevated specifically in Group 3 (Figure 17), more specifically in Group 3 alpha. The correlation analysis showed that the biological pathways most significantly correlated with *AANAT* expression were the phototransduction pathway (Table 4). Our analysis relates *AANAT* expression to numerous phototransduction genes over-expressed in the Group 3 subgroup of the Cavalli dataset. One interpretation, which has not been disproved, is that the cell type of origin of Group 3 alpha is a photoreceptor precursor cell expressing phototransduction genes and *AANAT*. This developmental pattern of expression also appears to be found in the neonatal rat pineal *AANAT* [85].


**Ubiquitin–Proteasome Pathway Regulation of Clock Genes in MB**


Figure 1 shows the role of ubiquitin ligases in the degradation of clock proteins, including F-box motif proteins that serve as adaptors for E3 ubiquitin ligase complexes. Several genes coding for ubiquitin ligases or adaptors are differentially expressed in MB at high significance.

*FBXL15* has the lowest expression in the SHH MB subgroup (Figure 12B). FBXL15, a substrate adaptor, plays a role in resetting the circadian clock in Drosophila by degrading the TIM protein and also contributes to the degradation of SMURF1 in the SHH pathway [86,87]. *FBXL21* shows one of the most significant differential expressions (Table 1). The FBXL21 protein binds to CRY proteins and opposes FBXL3’s action on CRY degradation [23]. It is implicated in the central circadian clock in mammals [22], although its classification as a pseudogene in humans remains debated.

*FBXW7* encodes a protein that regulates the degradation of REV-ERBα [28] and MYC [88,89]. Yeh et al. [90] discuss *FBXW7* as a tumor suppressor, with its under-expression found in MB (Figure 14) for most subtypes.

*MYCBP2*, a gene linked to the oncogene *MYC*, contributes to the degradation of NMNAT2 [91] and REV-ERBalpha [66].


**Summary of pathway analysis of clock gene correlates**


Table 4 summarizes the results of the KEGG pathway analysis of genes whose expression was most significantly correlated with that of clock genes. Table 4 lists the clock genes, the number of genes correlated with their expression, and the biological pathways over-represented in these lists of genes; the clock genes are listed in order of the *p* value for the most significant KEGG pathway. It is not a comprehensive study of all the biological pathways associated with clock genes.

The *ribosome* pathway was the most significant pathway associated with clock gene correlates, particularly in relation to copy number gain of chromosome 17q. We hypothesize that one or more clock genes regulate ribosome subunit gene expression at one or more stages in the ribosome biogenesis pathway. The significant correlations of *NR1D1* and *CIPC* expression with other genes (Table 4) suggest that REV-ERBα and CIPC may regulate transcription, including ribosome subunit genes. *FBXL21* and *MYCBP2*, which encode ubiquitin ligase components, could regulate the degradation of the clock proteins involved in feedback on the CLOCK/BMAL complex (Figure 1).

The *phototransduction* pathway was linked to the *RORA*, *USP2*, and *AANAT* correlates, with over-expression seen in MB Group 3 alpha (Table 4). Hooper et al. [55] showed the Group 3 MB expression profiles were similar to that of retinal rod cell precursors. We suggest the hypothesis that RORA may stimulate the transcription of phototransduction genes in the Group 3 alpha subtype.

Other significant KEGG pathways associated with clock gene correlates include the *GABAergic synapse*, *synaptic vesicle cycle*, *Fanconi anemia*, *WNT signaling*, *Hippo signaling*, and *peroxisome* pathways (Table 4), indicating potential mechanisms by which clock genes influence gene transcription in MB.


**Limitations of Transcription studies**


When analyzing and interpreting gene expression data, it is important to recognize the limitations of such analyses in addressing the full biological complexity of cancer. Predictions based on gene expression can identify potential drug targets and pathways; they can be validated in vitro in MB cell lines.

Another significant challenge lies in the reliance on single time-point data in many gene expression analyses, which fails to capture the dynamic nature of tumor evolution over time. This limitation makes it difficult to assess the long-term efficacy of potential treatments and to predict resistance mechanisms. Datasets incorporating longitudinal gene expression data —tracking changes over time—can greatly enhance the accuracy and reliability of predictions.

Furthermore, it is important to acknowledge that computational models are built upon biological assumptions and predefined pathways, which may not fully reflect the intricate and heterogeneous nature of cancer. An over-reliance on the existing knowledge can result in missed opportunities to uncover novel mechanisms and interactions. To overcome these limitations, machine learning approaches must be complemented with experimental validation and unbiased discovery techniques, ensuring that findings are both biologically and clinically meaningful.

## 5. Conclusions

We identified clock gene expression differences in public MB datasets, with major variations between the MB subgroups and subtypes. Aberrations in clock gene expression and feedback were group-specific, with major differences noted for *THRA*/*NR1D1* and *CIPC* loci. Key genes encoding proteins involved in clock protein degradation, including *FBXL21*, *FBXL3*, *FBXL15*, *MYCBP2*, and the deubiquitinase *USP2*, also showed differential expression.

Copy number gain of chromosome 17q was associated with increased clock gene expression, both for genes on 17q and on other chromosomes, suggesting that the isochromosome 17q syndrome significantly affects clock gene transcription in MB subgroups.

The ribosome pathway was the most significant KEGG pathway associated with clock gene expression, and it also correlated with survival, likely due to 17q copy number gain.

The pathway analysis of clock gene correlates also showed associations with the *phototransduction*, *GABAergic*, *synaptic vesicle*, *WNT signaling*, and *Fanconi anemia* pathways, with E-box binding being the most significant GO molecular term. MB should be considered a cancer in which E-box regulators play a role.

The Kaplan–Meier analysis identified the clock genes *CRY1*, *USP2*, *CLOCK*, *MYCBP2*, and *TIMELESS* as survival-related, with high expression of *CRY1*, *USP2*, *MYCBP2*, and *TIMELESS* linked to poor survival, and *CLOCK* to survival protection. These genes could serve as potential therapeutic targets in MB subgroups and subtypes. Methods targeting E-box regulators, including protein degradation strategies, should be explored for those currently lacking specific inhibitors.

## Figures and Tables

**Figure 1 cancers-17-00575-f001:**
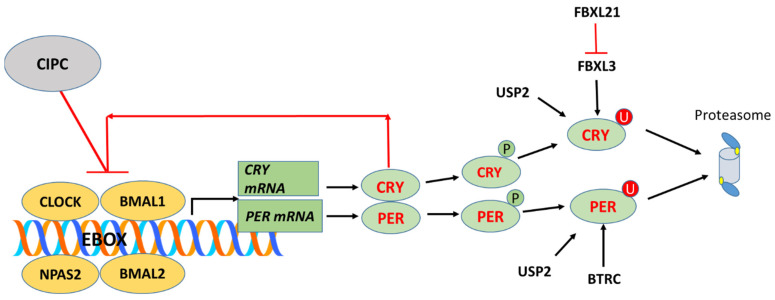
Core components of the circadian clock with transcription–translation feedback loop. Cry—cryptochrome; Per—period, BMAL1 (also known as ARNTL1)—Aryl Hydrocarbon Receptor Nuclear Translocator-Like Protein 1; BMAL2 (also known as ARNTL2)—Aryl Hydrocarbon Receptor Nuclear Translocator-Like Protein 2; CIPC—Clock Interacting Pacemaker; FBXL3—F-Box And Leucine-Rich Repeat Protein 3; FBXL21—F-Box And Leucine-Rich Repeat Protein 21; BTRC—Beta-Transducin Repeat-Containing E3 Ubiquitin Protein Ligase; USP2—Ubiquitin-Specific Peptidase 2. The degradation of CRY/PER proteins by the proteasome leads to decreased negative feedback.

**Figure 2 cancers-17-00575-f002:**
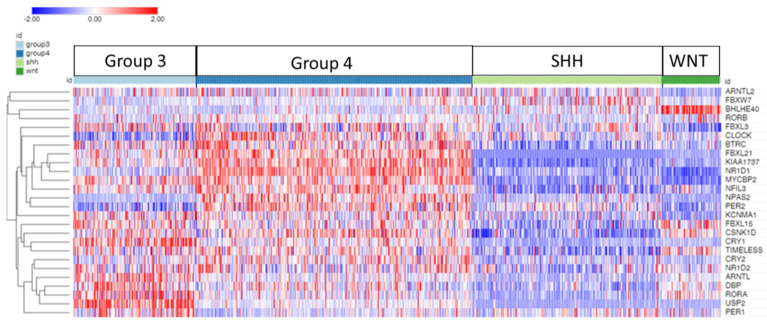
Heatmap and cluster analysis of the expression of clock genes in subgroups of MB. Each column is an individual, each row is a gene, and the color shows the low (blue) to high (red) gene expression levels. The 763 individuals were grouped by the MB molecular subgroups: Group 3, Group 4, SHH, and WNT. The differential expression of all the genes was significant at *p* < 0.001 except *RORB*; see Table 1.

**Figure 3 cancers-17-00575-f003:**
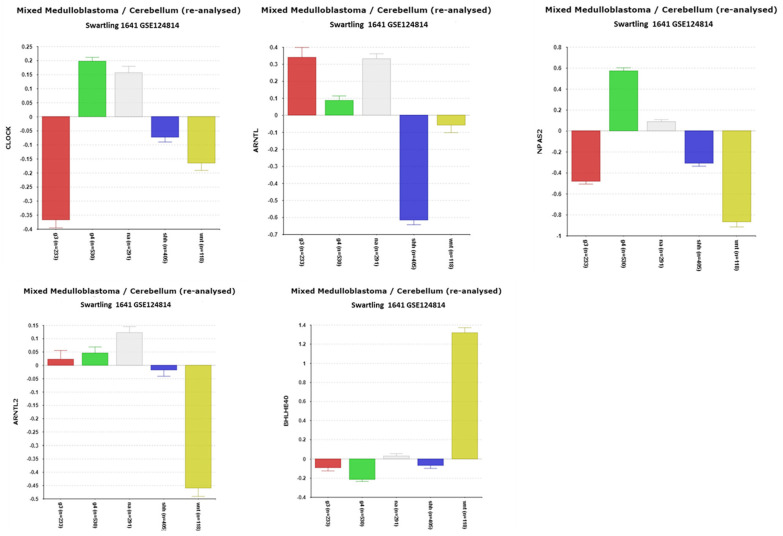
Differential expression of *CLOCK*, *NPAS2*, *ARNTL (BMAL1)*, *ARNTL2* (*BMAL2*), and *BHLHE40* in MB subgroups compared to expression in normal cerebellar (NA) tissue (Swartling dataset). Data expressed as relative gene expression by group. (*CLOCK*, F = 126.36, *p* = 1.32 × 10^−93^); (*NPAS2*, F = 283.34, *p* = 1.59 × 10^−183^); (*ARNTL*, F = 140.76, *p* = 6.58 × 10^−103^); (*ARNTL2*, F = 34.48, *p* = 1.26 × 10^−27^); (*BHLHE40*, F = 209.02, *p* = 7.20 × 10^−144^). Red—Group 3; Green—Group 4; Gray—NA (normal) in the Swartling dataset is the non-tumor group; blue—SHH; Yellow—WNT.

**Figure 4 cancers-17-00575-f004:**
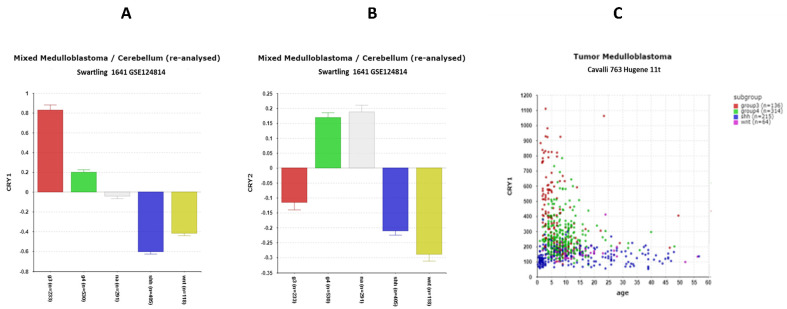
Cryptochrome expression in subgroups of medulloblastoma datasets. (**A**). CRY1 expression (F = 289.27, *p* = 1.69 × 10^−186^; (**B**). *CRY2* expression, F = 115.25, *p* =2.96 × 10^−86^). Red—Group 3; Green—Group 4; Gray—NA (normal) in the Swartling dataset is the non-tumor group. Blue—SHH; Yellow—WNT. (**C**). *CRY1* expression by age and MB subgroup.

**Figure 5 cancers-17-00575-f005:**
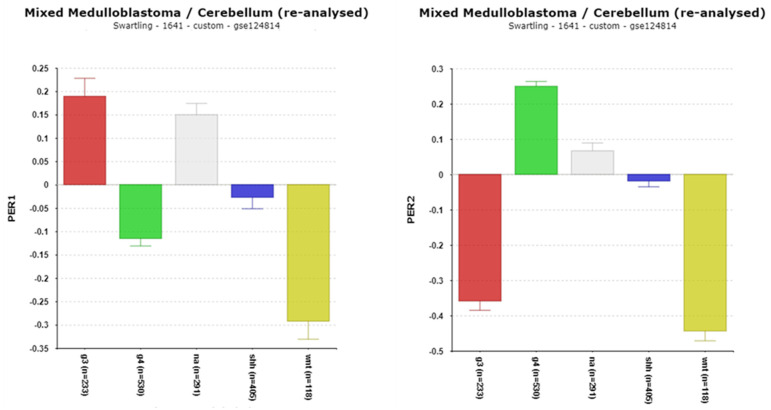
Differential expression of *PER1* and *PER2* in the Swartling dataset. *PER1*, F = 39.35, *p* = 1.9 3 × 10^−31^; *PER2*, F = 183.11, *p* = 6.79 × 10^−129^. Red—Group 3; Green—Group 4; Gray—NA (normal) in the Swartling dataset is the non-tumor group; blue—SHH; Yellow—WNT.

**Figure 6 cancers-17-00575-f006:**
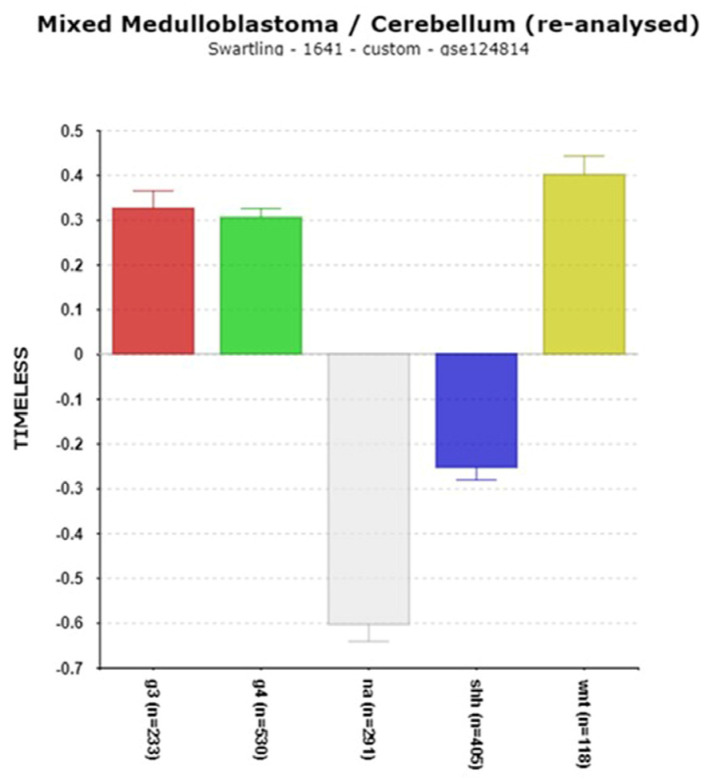
Differential expression of *TIMELESS* expression (F =188.16, *p* = 7.34 × 10^−132^). Red—Group 3; Green—Group 4; Gray—NA (normal) in the Swartling dataset is the non-tumor group; blue—SHH; Yellow—WNT.

**Figure 7 cancers-17-00575-f007:**
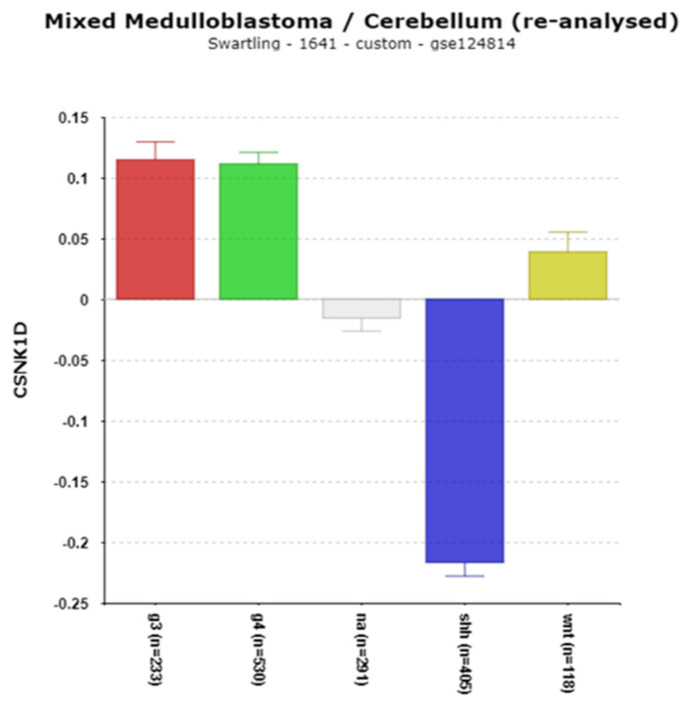
Differential expression of *CSNK1D*, F = 259.17, *p* = 191 × 10^−114^. Decreased expression in the SHH subgroup. Red—Group 3; Green—Group 4; Gray—NA (normal) in the Swartling dataset is the non-tumor group; blue—SHH; Yellow—WNT.

**Figure 8 cancers-17-00575-f008:**
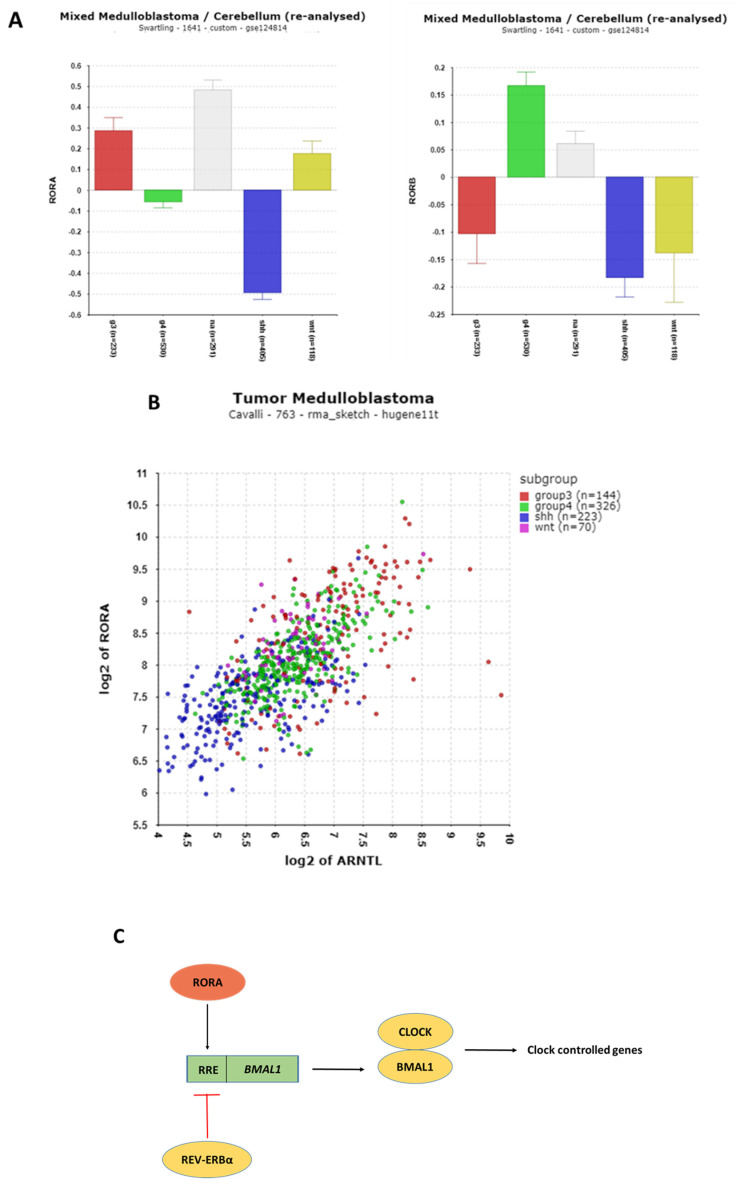
(**A**) Differential expression of *RORA* and *RORB* in the Swartling dataset. *RORA*, F = 85.84, *p* = 5.18 × 10^−66^; RORB, F = 19.47, *p* = 1.18 × 10^−15^. Red—Group 3; Green—Group 4; Gray—NA (normal) in the Swartling dataset is the non-tumor group; blue—SHH; Yellow—WNT. (**B**) Correlation of *RORA* and ARNTL (*BMAL1*) expression in the Cavalli dataset (r = 0.70, *p* = 1.19 × 10^−111^). MB subgroups identified by color. (**C**) RORA and REV-ERBα (NR1D1) proteins share binding sites on the *BMAL1* gene. RRE—Retinoic acid response element.

**Figure 9 cancers-17-00575-f009:**
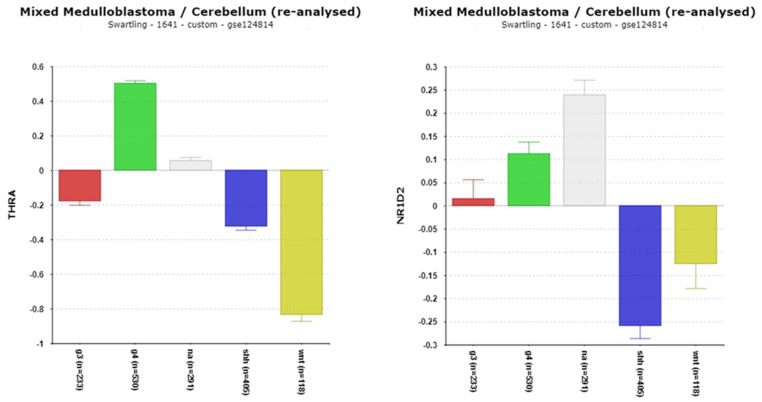
Differential expression of *THRA* (*THRA*/*NR1D1)* (F = 461.14, *p* = 4.34 × 10^−263^) and *NR1D2* (F = 40.21, *p* = 4.14 × 10^−32^) in the Swartling dataset. Note: THRA and NR1D1 are at the same locus on chromosome 17 (on opposite strands) as reported in the Cavalli dataset, the major source of the Swartling data. Red—Group 3; Green—Group 4; Gray—NA (normal) in the Swartling dataset is the non-tumor group; blue—SHH; Yellow—WNT.

**Figure 10 cancers-17-00575-f010:**
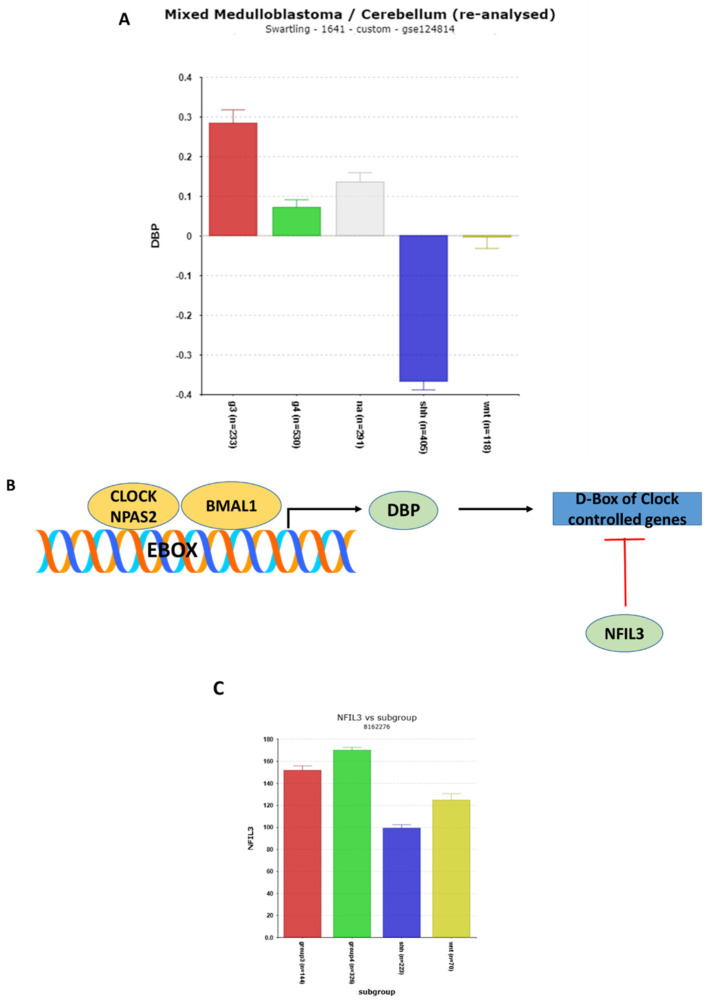
(**A**) *DBP* expression in MB subgroups. The depressed expression of *DBP* (F = 109.10, *p* = 4.06 × 10^−82^) in the MB SHH subgroup. Red—Group 3; Green—Group 4; Gray—NA (normal) in the Swartling dataset is the non-tumor group; blue—SHH; Yellow—WNT. (**B**) Opposite effects of DBP (D box Binding Protein) and NFIL3 (Nuclear Factor, Interleukin 3) on the D-box of clock genes. (**C**) Note expression for the *NFIL3* gene was not found in the Swartling dataset. Differential expression of *NFIL3* in MB subgroups in the Cavalli dataset was highly significant (F = 106.97, *p* = 9.10 × 10^−58^).

**Figure 11 cancers-17-00575-f011:**
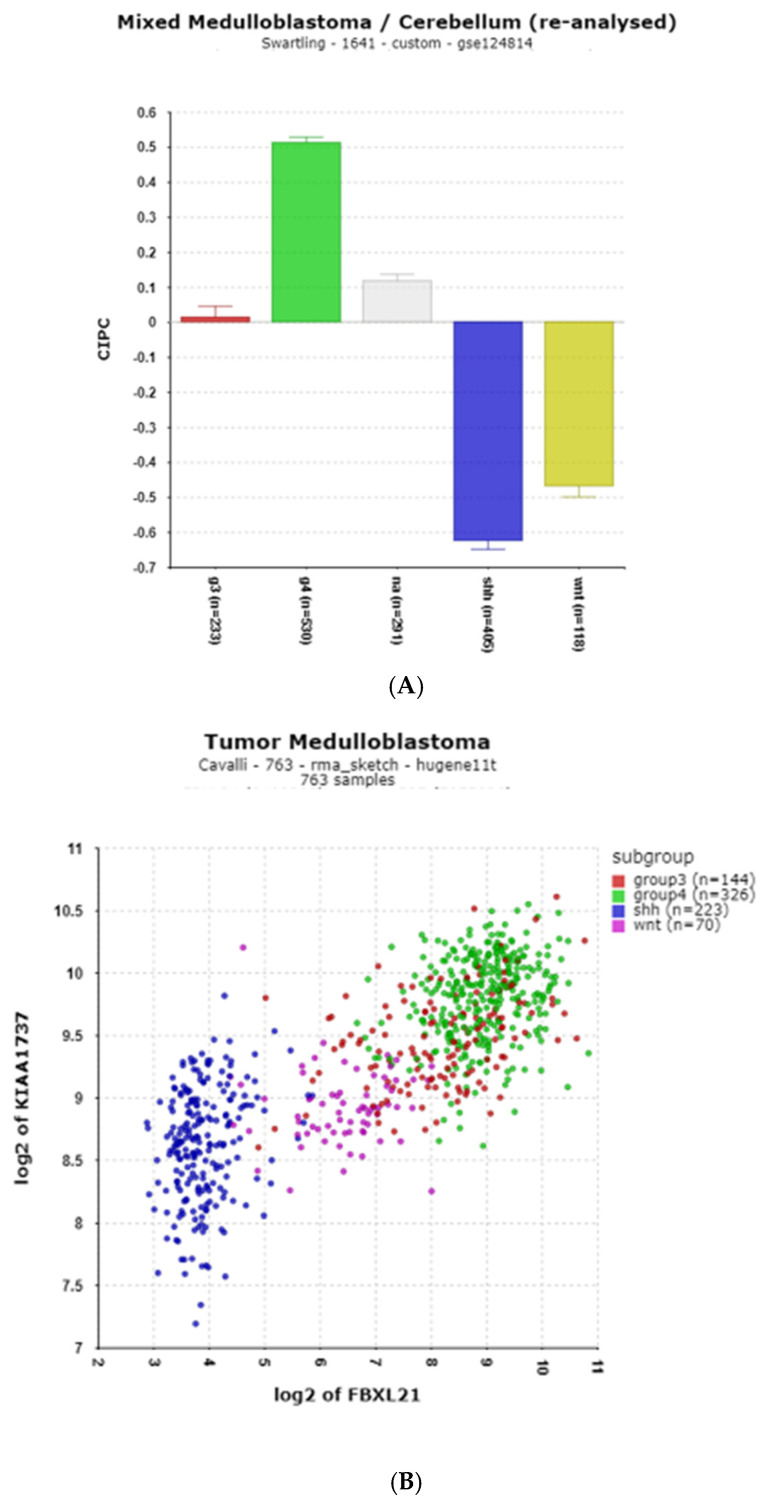
(**A**) Differential expression of *CIPC* (Clock Interacting Pacemaker) in MB subgroups. (F = 486.84, *p* = 3.39 × 10^−273^). Red—Group 3; Green—Group 4; Gray—NA (normal) in the Swartling dataset is the non-tumor group; blue—SHH; Yellow—WNT. (**B**) Correlation of *CIPC* and *FBXL21* (r = 0.790, *p* = 7.75 × 10^−164^).

**Figure 12 cancers-17-00575-f012:**
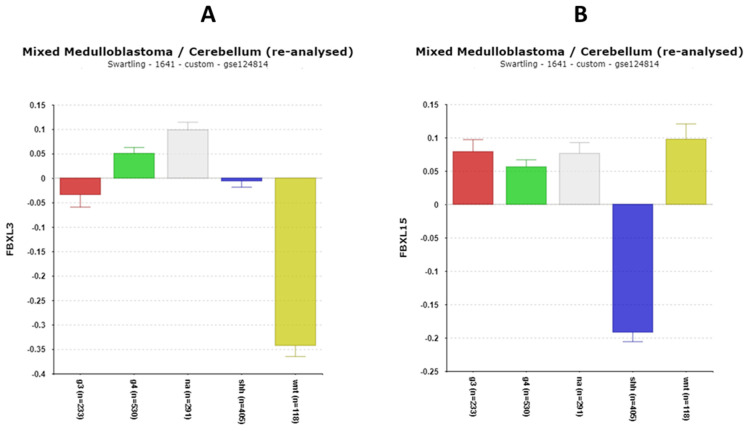
(**A**) Differential expression of *FBXL3* in the MB subgroups (F = 55.21, *p =* 1.31 × 10^−43^) by anova. Depression of *FBXL3* in the WNT subgroup compared to the NA group (by *t*-test, *p* = 1.63 × 10^−42)^). (**B**) Differential expression of *FBXL15* in the MB subgroups (F = 72.93, *p* = 9.80 × 10^−57^). Depression of *FBXL15* in the SHH subgroup compared to the NA group (by *t*-test, *p* = 1.06 × 10^−31^). Red—Group 3; Green—Group 4; Gray—NA (normal) in the Swartling dataset is the non-tumor group; blue—SHH; Yellow—WNT.

**Figure 13 cancers-17-00575-f013:**
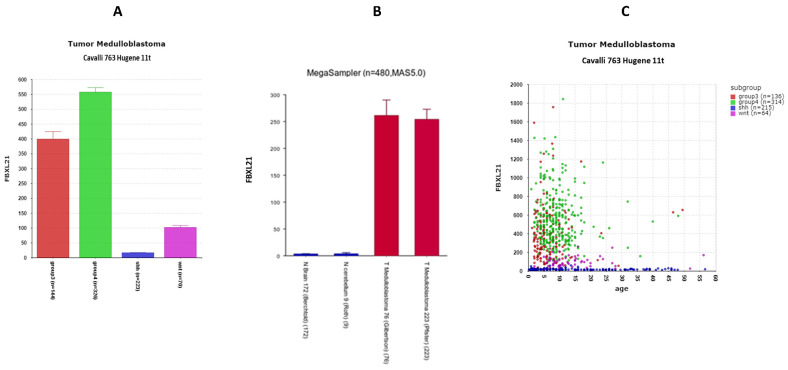
*FBXL21* expression in MB. (**A**) Expression of FBXL21 elevated in MB (Pfister dataset and Gilbertson dataset) compared to normal brain (Berchtold dataset) or normal cerebellum (Roth dataset), F = 51.49, *p* = 7.76 × 10^−29^. (**B**) Expression of FBXL21 elevated in MB Groups 3 and 4 MB compared to SHH and WNT MB, Cavalli dataset. F = 289.07, *p* = 4.14 × 10^−125^. (**C**) Age-related expression of *FBXL21* by subtype in Cavalli dataset.

**Figure 14 cancers-17-00575-f014:**
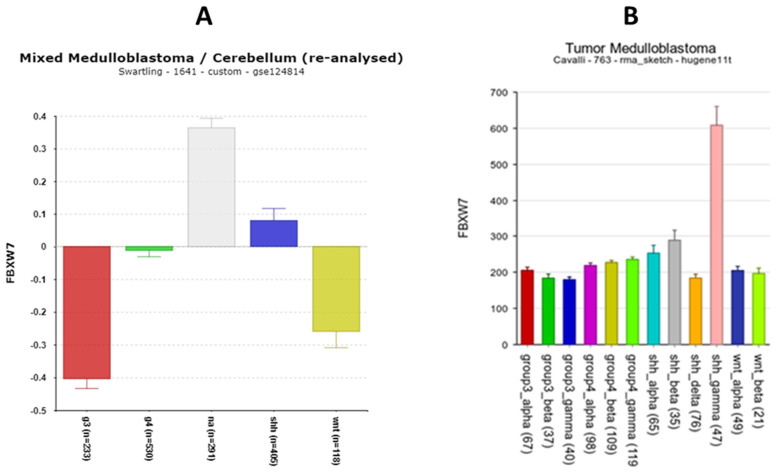
(**A**) Differential expression by subgroup of FBXW7 (F = 71.21, *p* = 1.74 × 10^−55^) in the Swarttling dataset. Red—Group 3; Green—Group 4; Gray—NA (normal) in the Swartling dataset is the non-tumor group; blue—SHH; Yellow—WNT. (**B**) Differential expression by 12 subtypes in Cavalli dataset (F = 40, *p* = 3.68 × 10^−60^). Increased expression in SHH gamma subtype.

**Figure 15 cancers-17-00575-f015:**
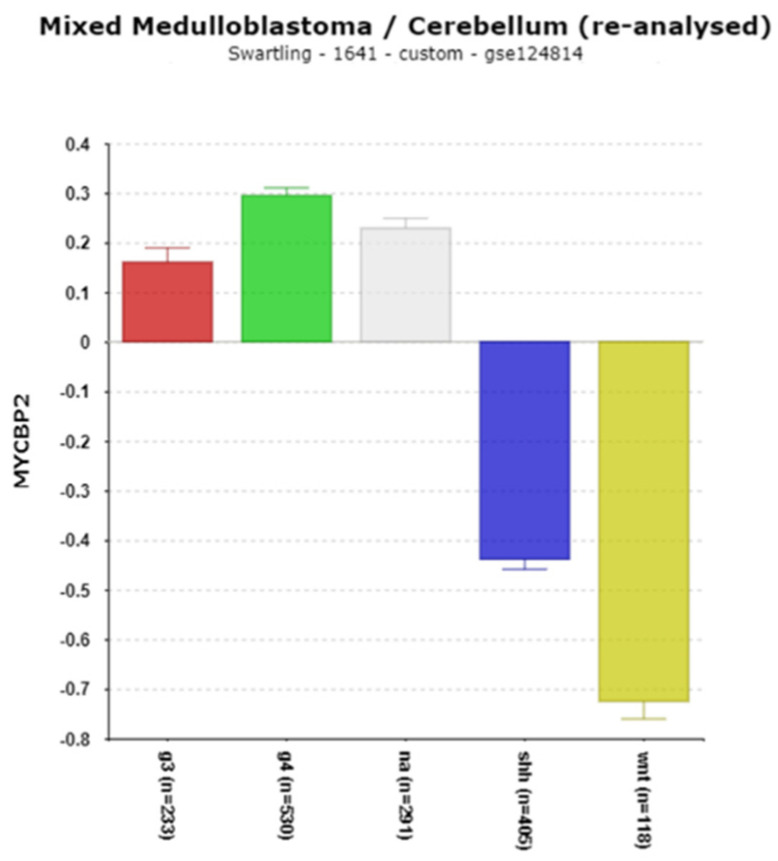
Differential expression by subgroup of *MYCBP2* (F = 350.75, *p* = 6.62 × 10^−216^). Red—Group 3; Green—Group 4; Gray—NA (normal) in the Swartling dataset is the non-tumor group; blue—SHH; Yellow—WNT.

**Figure 16 cancers-17-00575-f016:**
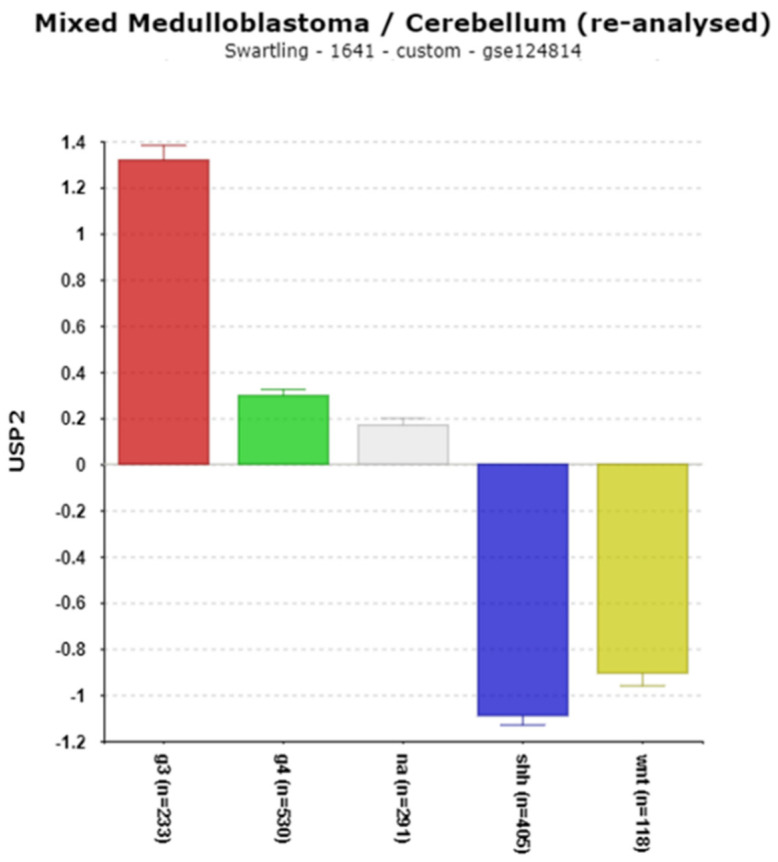
Differential expression of *USP2*. F = 508.57, *p* = 1.63 × 10^−281^. Red—Group 3; Green—Group 4; Gray—NA (normal) in the Swartling dataset is the non-tumor group; blue—SHH; Yellow—WNT.

**Figure 17 cancers-17-00575-f017:**
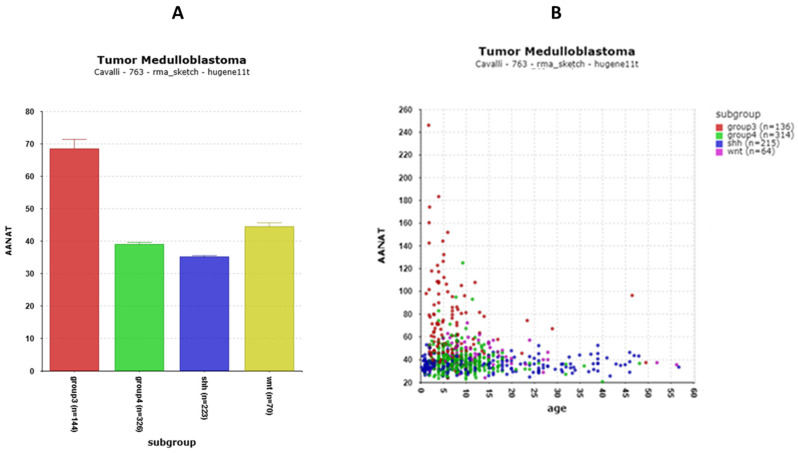
(**A**) *AANAT* expression is high in MB Group 3. F= 127.15.09, *p* = 9.93 × 10^−67^. (**B**) Age-related expression of *AANAT*.

**Figure 18 cancers-17-00575-f018:**
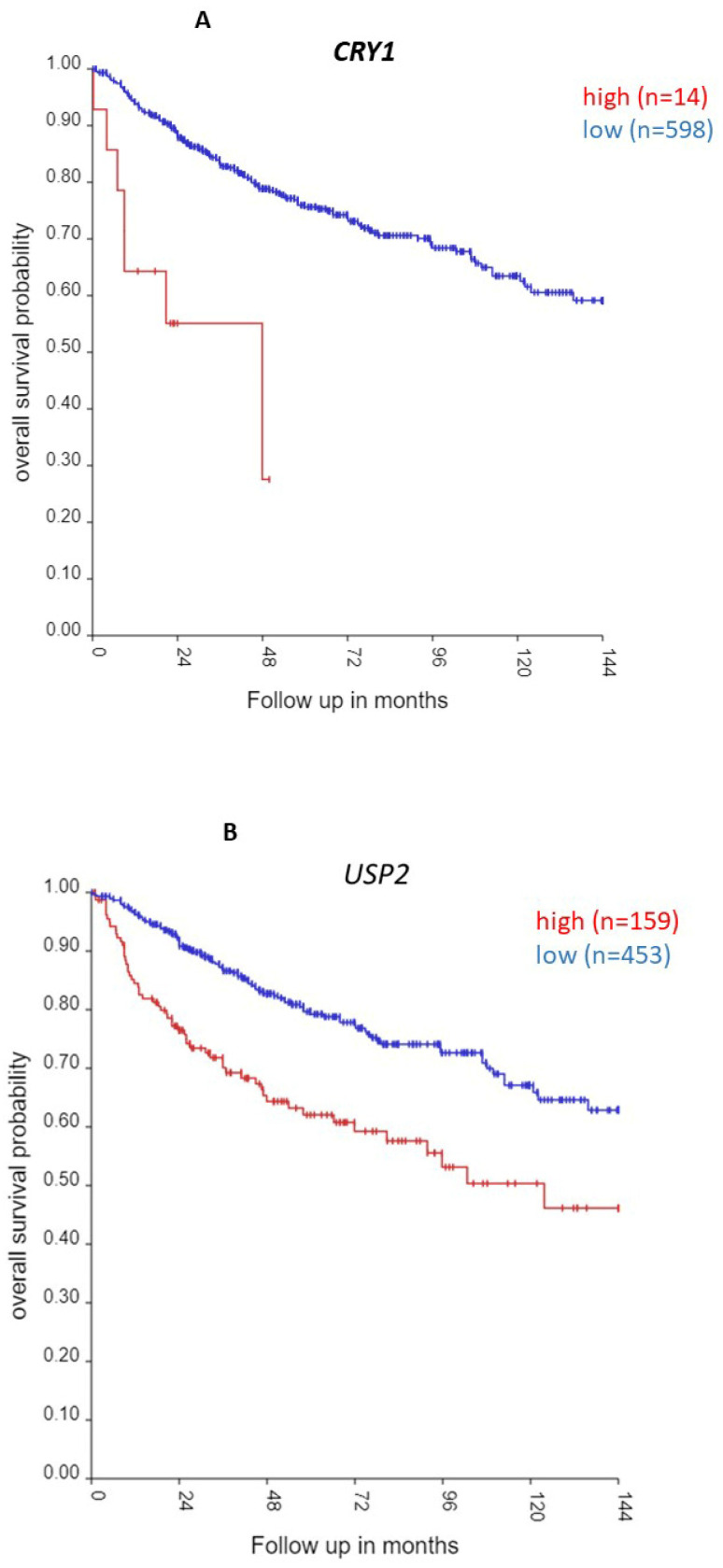
Most significant Kaplan–Meier Chi-squared *p* values for clock genes. (**A**). *CRY1*, Chi-squared = 22.33, *p* = 2.29 × 10^−6^; (**B**). *USP2*, Chi-squared = 21.73, *p* = 3.41 × 10^−6^; (**C**). *CLOCK*, Chi-squared = 20.21, *p* = 6.96 × 10^−6^. Red—survival curve for high gene expression; blue—survival curve for low gene expression. The factors associated with high expression of *CRY1* include subgroup (*p* = 1.29 × 10^−80^), age group (*p* = 2.47 × 10^−10^), metastases status (*p* = 4.04 × 10^−5^), and sex (*p* = 1.49 × 10^−3^). The factors associated with high expression of *USP2* include subgroup (*p* = 1.57 × 10^−119^), age group (*p* = 1.01 × 10^−8^), and metastases status (*p* = 6.67 × 10^−3^). The factors associated with *CLOCK* expression include subgroup (*p* = 1.81 × 10^−41^) and age group (*p* = 1.83 × 10^−8^).

**Table 1 cancers-17-00575-t001:** Clock genes: differential expression in MB subgroups by analysis of variance.

‘CLOCK’ Genes	F (Anova) Cavalli 4 Groups	*p*	*p* (fdr)	F (Anova) Swartling 4 MB Groups Plus NT	*p*	*p* (fdr)
*THRA*/*NR1D1*	423.43	1.43 × 10^−161^	Nc	461.14	4.34 × 10^−263^	3.47 × 10^−262^
*CIPC (KIAA1737)*	379.47	1.67 × 10^−150^	4.17 × 10^−149^	486.84	3.39 × 10^−273^	4.07 × 10^−272^
*FBXL21*	289.07	4.14 × 10^−125^	5.17 × 10^−124^	Not found *		
*USP2*	271.01	1.57 × 10^−119^	1.31 × 10^−118^	508.57	1.63 × 10^−281^	3.90 × 10^−280^
*MYCBP2 (PAM)*	231.63	1.13 × 10^−106^	7.08 × 10^−106^	350.75	6..62 × 10^−216^	3.97 × 10^−215^
*CRY1*	160.65	1.29 × 10^−80^	6.43 × 10^−80^	289.27	1.69 × 10^−186^	8.11 × 10^−186^
*BHLHE40*	139.4	6.08 × 10^−72^	2.54 × 10^−71^	209.02	7.20 × 10^−144^	2.47 × 10^−143^
*CSNK1D*	112.9	1.88 × 10^−60^	6.72 × 10^−60^	159.17	1.91 × 10^−114^	4.18 × 10^−114^
*NFIL3*	106.98	9.10 × 10^−58^	2.84 × 10^−57^	Not found *		
*PER2*	101.21	4.08 × 10^−55^	1.13 × 10^−54^	118.11	6.79 × 10^−129^	1.81 × 10^−128^
*NPAS2*	92.82	3.53 × 10^−51^	8.82 × 10^−51^	283.34	1.59 × 10^−−83^	6.35 × 10^−183^
*BTRC*	83.47	1.11 × 10^−46^	2.53 × 10^−46^	164.35	1.26 × 10^−117^	3.03 × 10^−117^
*RORA*	78.39	3.52 × 10^−44^	7.34 × 10^−44^	85.84	5.18 × 10^−66^	7.31 × 10^−66^
*CLOCK*	77.07	1.59 × 10^−43^	3.06 × 10^−43^	126.36	1.32 × 10^−93^	2.43 × 10^−93^
*TIMELESS*	67.05	1.82 × 10^−38^	3.24 × 10^−38^	188.16	7.34 × 10^−132^	2.20 × 10^−131^
*ARNTL (BMAL1)*	59.29	1.91 × 10^−34^	3.18 × 10^−34^	140.76	6.58 × 10^−103^	1.32 × 10^−102^
*FBXL15*	57.12	2.67 × 10^−33^	4.18 × 10^−33^	72.93	9.80 × 10^−57^	1.31 × 10^−56^
*DBP*	55.93	1.14 × 10^−32^	1.67 × 10^−32^	109.1	4.06 × 10^−82^	6.5 × 10^−82^
*KCNMA1*	36.6	4.23 × 10^−22^	5.88 × 10^−22^	Not found *		
*CRY2*	35.25	2.48 × 10^−21^	3.26 × 10^−21^	115.25	2.96 × 10^−86^	5.07 × 10^−86^
*PER3*	34.22	9.45 × 10^−21^	Nc	100.54	2.81 × 10^−76^	4.22 × 10^−76^
*FBXL3*	27.84	4.37 × 10^−17^	5.47 × 10^−17^	55.21	1.31 × 10^−43^	1.57 × 10^−43^
*FBXW7*	22.21	8.55 × 10^−14^	1.02 × 10^−13^	71.21	1.74 × 10^−55^	2.20 × 10^−55^
*NR1D2*	19.64	2.87 × 10^−12^	3.26 × 10^−12^	40.21	4.14 × 10^−32^	4.73 × 10^−32^
*PER1*	18.63	1.15 × 10^−11^	1.25 × 10^−11^	39.35	1.93 × 10^−31^	2.11 × 10^−31^
*ARNTL2 (BMAL2)*	12.57	5.01 × 10^−8^	5.22 × 10^−8^	34.48	1.26 × 10^−27^	1.31 × 10^−27^
*RORC*	9.86	2.20 × 10^−6^	Nc	Not found		
*RORB*	2.57	0.053	ns	19.47	1.18 × 10^−15^	1.18 × 10^−15^

* not found in the Swartling dataset. Nc—not calculated because the reporter codes for more than one gene.

**Table 2 cancers-17-00575-t002:** Survival analysis of clock-related genes; Kaplan–Meier and Cox hazard ratios.

Survival-Related ‘CLOCK’ Genes	Chi-Squared	Kaplan–Meier *p* Values		HR	Hazard Ratio *p* Values
*CRY1*	22.33	2.29 × 10^−6^	high worse	1.4	0.00094
*USP2*	21.73	3.14 × 10^−6^	high worse	1.3	0.000035
*CLOCK*	20.21	6.96 × 10^−6^	low worse	0.61	0.024
*MYCBP2 (PAM)*	18.29	1.90 × 10^−5^	high worse	1.5	0.018
*TIMELESS*	15.23	9.53 × 10^−5^	high worse		ns
*UTS3*/*PER3* **	14.83	1.18 × 10^−4^	low worse	0.75	0.013
*FBXL21*	13.63	2.23 × 10^−4^	high worse		ns
*BTRC*	13.1	2.96 × 10^−4^	low worse		ns
*ARNTL2 (BMAL2)*	12.91	3.27 × 10^−4^	high worse	1.5	0.0035
*CSNK1D*	11.5	6.95 × 10^−4^	high worse	2.2	0.0084
*BHLHE40*	10.65	1.10 × 10^−3^	low worse		ns
*NR1D2*	10.62	1.12 × 10^−3^	low worse		ns
*THRA*/*NR1D1* **	9.71	1.84 × 10^−3^	high worse		ns
*RORA*	8.58	3.40 × 10^−3^	high worse		ns
*RORB*	8.39	3.78 × 10^−3^	high worse	1.2	0.03
*PER1*	8.25	4.08 × 10^−3^	low worse		ns
*KCNMA1*	6.96	8.35 × 10^−3^	high worse		ns
*PER2*	6.83	8.96 × 10^−3^	high worse		ns
*NFIL3*	6.67	9.80 × 10^−3^	high worse	1.3	0.034
*CRY2*	6.15	1.3 × 10^−2^	high worse		ns
*DBP*	6.00	1.40 × 10^−2^	high worse		ns
*FBXW7*	5.76	1.60 × 10^−2^	low worse		ns
*NPAS2*	5.12	2.40 × 10^−2^	high worse		ns
*AANAT*	4..95	2.60 × 10^−2^	high worse		ns
*ARNTL (BMAL1)*	4.00	4.50 × 10^−2^	low worse		ns
*CIPC (KIAA1737)*	3.99	4.60 ×10^−2^	high worse		ns
*FBXL15*	3.98	4.60 × 10^−2^	low worse		ns
*LINGO4*/*RORC* **	3.16	0.76 × 10^−1^	ns		ns
*FBXL3*	2.89	8.90 × 10^−2^	ns		ns

** same reporter listed for two genes.

**Table 3 cancers-17-00575-t003:** Clock genes: differential expression related to copy number gain of chromosome 17q.

‘CLOCK’ Genes	Chromosomal Location	Means ± s.e. Normal	Means ± s.e. Gain 17q	*p* Value for *t*-Test
*THRA*/*NR1D1*	17q21.1	921.99 ± 16.44	1533.30 ± 22.89	1.57 × 10^−84^
*CIPC (KIAA1737)*	14q24.3	551.55 ± 10.71	891.27 ± 13.19	7.35 × 10^−73^
*CSNK1D*	17q25.3	647.89 ± 5.67	793.71 ± 5.70	9.19 × 10^−60^
*FBXL21*	5q31.1	178.26 ± 12.47	525.69 ± 16.53	1.09 × 10^−55^
*MYCBP2*/*PAM*	13q22.3	530.72 ± 10.14	693.05 ± 8.66	2.08 × 10^−29^
*PER2*	2q37.3	212.53 ± 3.20	273.45 ± 4.29	8.03 × 10^−29^
*NPAS2*	2q11.2	188.29 ± 8.87	355.41 ± 12.84	2.41 × 10^−26^
*NFIL3*	9q22.31	123.99 ± 2.75	165.33 ± 2.42	6.50 × 10^−26^
*CRY2*	11p11.2	193.97 ± 2.33	236.28 ± 4.44	2.64 × 10^−18^
*BTRC*	10q24.32	329.77 ± 4.98	390.03 ± 5.61	3.80 × 10^−15^
*CLOCK*	4q12	425.20 ± 4.43	486.90 ± 6.79	1.06 × 10^−14^
*TIMELESS*	12q13.3	159.39 ± 3.22	189.19 ± 2.82	3.06 × 10^−11^
*BHLHE40*	3p26.1	250.49 ± 10.80	164.57 ± 4.88	7.11 × 10^−11^
*DBP*	19q13.33	189.32 ± 3.99	230.81 ± 5.81	2.07 × 10^−9^
*PER1*	17p13.1	137.78 ± 4.05	111.29 ± 2.51	2.67 × 10^−7^
*CRY1*	12q23.3	224.89 ± 8.22	280.67 ± 8.54	3.52 × 10^−6^
*NR1D2*	3p24.2	422.73 ± 6.49	463.72 ± 8.49	1.02 × 10^−4^
*USP2*	11q23.3	276.55 ± 18.90	373.71 ± 14.59	1.11 × 10^−4^
*FBXL15*	10q24.32	47.39 ± 0.46	49.86 ±0.44	1.62 × 10^−4^
*FBXW7*	4q31.3	260.54 ± 9.77	222.39 ± 3.97	1.08 × 10^−3^
*FBXL3*	13q22.3	387.40 ± 4.54	409.41 ± 5.47	1.88 × 10^−3^
*ARNTL2*	12p11.23	59.71 ± 1.30	64.99 ± 1.70	1.23 × 10^−2^
*KCNMA1*	10q22.3	288.29 ± 11.66	326.16 ± 10.74	2.02 × 10^−2^
*AANAT*	17q25.1	45.16 ± 1.35	42.35 ± 0.94	8.14 × 10^−2^ ns
*ARNTL*	11p15.3	96.57 ± 4.40	89.08 ± 2.77	1.78 × 10^−1^ ns
*RORA*	15q22.2	283.04 ± 8.28	299.43 ± 8.91	1.81 × 10^−1^ ns
*RORB*	9q21.13	61.65 ± 3.43	60.31 ± 3.03	7.76 × 10^−1^ ns

**Table 4 cancers-17-00575-t004:** Pathway analysis of clock genes.

CLOCK Genes	Number of Correlates r > 0.50	Most Significant KEGG Pathways Over-Represented	# of Genes Representing Pathway	*p* Value for Pathway
*THRA*/*NR1D1*	1041	ribosome	59	1.44 × 10^−80^
*CIPC (KIAA1737)*	1063	ribosome	57	1.34 × 10^−73^
*FBXL21*	605	ribosome	44	2.32 × 10^−74^
*MYCBP2*	423	ribosome	29	1.34 × 10^−48^
*UTS2*/*PER3*	18	circadian rhythm	3	2.62 × 10^−43^
*RORA*	184	phototransduction	7	2.30 × 10^−33^
*NPAS2*	85	synaptic vesicle cycle	6	2.09 × 10^−21^
*CRY2*	52	GABAergic synapse	5	4.38 × 10^−19^
*TIMELESS*	31	Fanconi anemia	3	1.07 × 10^−18^
*BTRC*	34	Oocyte meiosis	4	5.61 × 10^−12^
*USP2*	161	phototransduction	4	1.74 × 10^−11^
*AANAT*	157	phototransduction	4	1.74 × 10^−11^
*BHLHE40*	148	WNT signaling	10	2.47 × 10^−9^
*DBP*	62	GABAergic synapse	4	3.98 × 10^−9^
*FBXW7*	88	morphine addiction	5	3.98 × 10^−8^
*NFIL3*	82	Hippo signaling	5	3.28 × 10^−6^
*CSNK1D*	105	prolactin signaling	4	3.45 × 10^−6^
*CRY1*	139	peroxisome	4	1.12 × 10^−3^
*PER2*	22	circadian rhythm	1	2.13 × 10^−3^
*FBXL3*	37	circadian rhythm	1	4.90 × 10^−2^
*ARNTL (BMAL1)*	8	*		
*ARNTL2 (BMAL2)*	0	*		
*PER1*	5	*		
*RORB*	0	*		
*LINGO4*/*RORC*	0	*		
*FBXL15*	0	*		

* no KEGG pathway detected at a cutoff of r > 0.50.

## Data Availability

The data referred to in this manuscript are publicly available (GEO ID: GSE124814) and at the R2 Genomics Analysis and Visualization Platform (http://r2:amc.nl, last access date, 12 December 2024) and are available upon reasonable request from the corresponding author.

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
