# Peer review of "Transcription of Clock Genes in Medulloblastoma"

_cancers, 2025, doi:10.3390/cancers17040575_

Round 1

Reviewer 1 Report

Comments and Suggestions for Authors

This manuscript explores the expression of circadian clock genes in medulloblastoma (MB) subgroups using publicly available gene expression datasets. The research provides insights into subgroup-specific variations and their prognostic significance, particularly for clock-related genes. The study is scientifically interesting and relevant, given the emerging role of circadian rhythms in cancer biology. However, several methodological and presentation issues need attention to improve clarity, rigor, and overall impact.

Major Points

1.  The introduction provides an exhaustive review of circadian clock biology, but it lacks a succinct summary of the study’s hypothesis and objectives. I suggest the authrıus to condense the background and explicitly outline the study goals to guide the reader.

2. The methodology for batch correction and dataset integration remains underexplained. A detailed description of how batch effects were managed, especially for comparing tumor vs. normal tissue, is essential for reproducibility.

3. While ANOVA and survival analyses were performed, there is limited discussion on adjustments for multiple testing, which is critical given the large number of genes analyzed. Include details on whether false discovery rate (FDR) correction or other methods were applied.

4. The authors identify CRY1, USP2, and CLOCK as key survival-associated genes. While these findings are significant, the lack of experimental validation limits their translational potential. Consider discussing how these findings might be confirmed experimentally.

5. The discussion highlights potential therapeutic targets but does not sufficiently address the limitations of relying solely on computational analyses for drug discovery. A balanced discussion is necessary to contextualize these findings.

Minor Points

1. The heatmaps and Kaplan-Meier survival plots could be better annotated. For example, labeling specific subgroup features (e.g., age, gender) within the survival plots might enhance interpretability.

2. While the manuscript is well-written overall, certain sections (e.g., Methods) are verbose and could be streamlined for clarity.

3. The manuscript does not address ethical compliance related to the use of public datasets. Clarify adherence to ethical guidelines.

Author Response

  1.   The introduction provides an exhaustive review of circadian clock biology, but it lacks a succinct summary of the study’s hypothesis and objectives…

Response: We have added a summary of the hypothesis:

The hypothesis for this study is that clock genes are differentially expressed in the four major MB subgroups which may have an impact on clinical outcome and therapeutic potential.  Our findings show major variations in clock genes are related to the isochromosome 17 aberration.  Furthermore our data show that variations in several clock genes are associated with survival.  

  1. Comment: The methodology of batch correction and dataset integration remains underexplained.  A detailed description of how batch effects were managed, especially for comparing tumor vs. normal tissue, is essential for reproducibility. 

                Response:           I have added a sentence referring the reader to the Weishaupt paper which describes in detail the methods of batch correction and dataset integration.    

“A detailed description of batch correction and dataset integration used in producing the Swartling dataset  is found in Weishaupt et al. “. 

                Having said this I acknowledge that the NT data has limitations as controls for tumor data. 

  1. Comment:  While anova and survival analysis were performed, there is limited discussion on adjustments for multiple testing, which is critical given the large number of genes analyzed…

Response:  We have added the p values with false discovery rate in Table 1.  There was very little difference in the p values and the differences are so small that they do not affect any of our interpretations of the data.

  1. Comment: The authors identify CRY1, USP2, and Clock as key survival-associated genes. While these findings are significant, the lack of experimental validation limits their translational potential.  Consider discussing how these findings might be confirmed experimentally.

                Response.  We have added to the discussion the following  (page 26);

These findings can be validated through gene expression analysis using qRT-PCR and RNA-seq, followed by assessing protein levels using standard western blot technique or more advance proteomics assays. Additionally, functional studies should be conducted using medulloblastoma cell lines to further investigate the impact of these findings. The observed upregulation of CRY1 and USP2 may contribute to increased CRY1 stabilization, leading to enhanced DNA damage repair and inhibition of apoptosis. Evaluating the effects of these alterations on cell proliferation and migration will provide valuable in-sights into their role in tumor progression. Furthermore, drug sensitivity assays will be crucial in exploring potential therapeutic approaches by testing USP2 inhibitors and assessing their effects on cell viability, apoptosis (via Annexin V/PI staining), and tumorigenic properties.

  1. Comment: The discussion highlights potential therapeutic targets but does not sufficiently address the limitations of relying solely on computational analysis for drug discovery. A balanced discussion is necessary to contexualize these findings. 

                Response:  I agree with the reviewer.  We definitely do not want to give the impression that we can rely solely on statistical analysis of gene transcription for drug discovery.  It is only one part of the process.  We have added the following to the discussion:

When analyzing and interpreting gene expression data, it is important to recognize the limitations of  such analyses in addressing the full biological complexity of cancer. Predictions based on gene expression can identify potential drug targets and pathways; they can be validated in vitro in MB cell lines and in mouse models of MB. 

Another significant challenge lies in the reliance on single time-point data in many gene expression analyses, which fails to capture the dynamic nature of tumor evolution over time. This limitation makes it difficult to  assess the long-term efficacy of potential treatments and to predict resistance mechanisms. Datasets incorporating longitudinal gene expression data —tracking changes over time—can greatly enhance the accuracy and reliability of predictions.

Furthermore, it is important to acknowledge that computational models are built up-on biological assumptions and predefined pathways, which may not fully reflect the intri-cate and heterogeneous nature of cancer. An over-reliance on existing knowledge can re-sult in missed opportunities to uncover novel mechanisms and interactions. To overcome these limitations, machine learning approaches must be complemented with experimental validation and unbiased discovery techniques, ensuring that findings are both biological-ly and clinically meaningful.     

Minor points

  1. Comment: The heatmaps and Kaplan-Meier survival plots could be better annotated…

Response:  I have added to the legend (Fig 2) of the heatmap as follows:

The 763 individuals were grouped by the MB molecular subgroups, Group 3, Group 4, SHH, and WNT . Differential expression of all genes were significant at p < 0.001 except RORB, see Table 1. 

Response:  I could not find a way to incorporate subgroup features directly within the survival plots.  However I have added to the legends of the survival plots indicating that subgroup, age, gender and metastases status significantly influence gene expression.  I would expect these factors to contribute to overall survival.

  1. While the manuscript is well-written overall, certain sections (e.g Methods) are verbose...

Response:  the Methods section is actually very short.  I did edit it for redundancy.   However, I added information on the source of the samples in the Cavalli and Swartling studies and the details of approval by institutional review boards (see next comment).

  1. Comment: The manuscript does not address ethical compliance related to the use of public datasets.  Clarify adherence to ethical guidelines. 

Response:   The authors of the Cavalli dataset, the major dataset used in the present study, made their data publicly available through the R2 Genomics Analysis and Visualization platform so that the data would be readily available to all researchers.  I have therefore no ethical issues in using their data for our publication on clock genes.  The Swartling dataset (a meta-analysis including the Cavalli dataset) was also made available through the R2 Genomics platform.

Ethical issues of concern would include whether the original projects were approved by the institutional review boards of the various institutions at which the tissues were collected.  The Cavalli paper states “All medulloblastoma samples were collected at diagnosis after obtaining informed consent from subjects as part of the Medulloblastoma Advanced Genomics International Consortium.”  In the methods of the Cavalli paper the authors list the institutions from which Approval was obtained from institutional research ethics boards for collection of tissue samples. 

I have added two sentences describing the source of the subjects and cite the Cavalli paper as a source of the list of institutions from which approval was obtained.

 The medulloblastoma samples of the Cavalli dataset were collected with informed consent as part of the Medulloblastoma Advanced Genomics International Consortium and approval from institutional review boards of the various institutions [40].

Reviewer 2 Report

Comments and Suggestions for Authors

I am ready with my reviews for manuscript titled “Transcription of Clock genes in Medulloblastoma” by Vriend and Glogowska. In the present study the authors have performed a descriptive analysis of the differential expression of clock gene in different subgroups of medulloblastoma (MB). They have studied the publicly available Cavalli and Swartling (reanalysis of several datasets) MB datasets to investigate the transcription of circadian clock genes. Additionally, they have also performed the survival and pathway analysis based on the expression of clock-related genes. This study provides several crucial pieces of information that can be further investigated to understand the role of clock-related genes in the onset and progression of MB. However, the current form of the manuscript needs major corrections before its final acceptance.

1.       Authors should describe more about the heatmap shown in Fig. 2. How the expression of these clock-related genes looks in the different MB subgroup.

2.       Please shuffle the position ARNTL and NPAS2 expression data in figure 3. As ARNTL is described first in the text.

3.       Which data support the following statement. Please provide figure or table number.

-          “Among the 12 MB subtypes of the 4 subgroups, highest expression of CRY1 was found in the Cavalli Group 3 alpha subtype (p = 2.77 x 10-56 vs NT)”- Page 8.

-          “Our analysis found that high expression of this gene was statistically associated with copy number gain of chromosome 17q; its expression was elevated in individuals with 17q copy number gain” – Page 9.

-          “Seven genes of this pathway—CNGB1, GNAT1, GRK1, GUCY2D, PDE6B, PDE6G, and RCVRN—were highly correlated with RORA (r > 0.50). These genes were markedly over-expressed in the MB Group 3 alpha subtype in the Cavalli dataset”- Page 10.

-          “The refined classification of Group 3 MBs into subtypes—Group 3 alpha, beta, and gamma—in the Cavalli dataset highlights increased expression of phototransduction genes in the Group 3 alpha subtype” - Page 10.

-          “Similarly, in the Cavalli dataset, NFIL3 expression was lowest in the SHH group com-pared to the other three subgroups (F = 106.98, p = 9.10 x 10-58)” -Page 13.

4.       Please correct the formatting of Fig. 8 legend.

5.       For NR1D2, the most significant difference by subgroup was observed as a depression in expression in the SHH subgroup compared to normal tissue (p = 3.86 x 10.².) (Fig. 8)- Page 12. Its Fig. 9 instead of Fig. 8.

6.       In the manuscript, many figures contain subheadings, while others are missing them. Please maintain a consistency.

7.       There is repetition of data in Fig. 13.

8.       “THRA/NR1D1, located on 17q, is elevated in individuals with 17q copy number gain, as shown in Table 3, along with CIPC, FBXL21, and MYCBP2”- Page 24. Please rewrite this sentence.

Author Response

  1. Comment Authors should describe  more about the heatmap shown in Fig. 2.  How the expression of these clock-related genes looks in the different MB subgroups.

        Response:  I have added a sentence as follows: The heatmap illustrates the differential expression of the clock genes by subgroup.  Highest expression was found mainly in Groups 3 and 4, but depended on individual gene expression as shown in the figures of individual gene expression below. 

  1. Comment:  Please shuffle the position ARTNL and NPAS2 expression data in figure 3. 

                Response:  Done

  1. Comment: Which data support the following statements.  Please provide figure or table number.

“Among the 12 MB subtypes of the 4 subgroups, highest expression of CRY1 was found in the Cavallli Group 3 alpha subtype (p = 2.77, x 10-56 vs NT)”.  Page 8

Response:  I have revised the sentence as follows so that it refers to Figure 4A  “Among the 4 MB subgroups, highest expression of CRY1 was found in the Group 3 MBs (Figure 4A) (p = 2.77 x 10-56 vs the NT group)

“Our analysis found that high expression of this gene was statistically associated with copy number gain of chromosome 17q; its expression was elevated in individuals with 17q copy number gain” page 9

Response:  I added Table 3 to the sentence. 

“Seven genes of this pathway—CNGB1, GNAT1, GRK1, GUCY2D, PDE6B, PDE6G, and RCVRN—were highly correlated with RORA (r > 0.50). These genes were markedly over-expressed in the MB Group 3 alpha subtype in the Cavalli dataset” page 10

Response:  I have added a supplemental file for the phototransduction genes and refer to it here.

“The refined classification of Group 3 MBs into subtypes-Group 3 alpha, beta, and gamma-in the Cavalli dataset highlights increased expression of phototransduction genes in the Group 3 alpha subtype.”  Page 10

Response:  Again I have referred to the supplemental file for these genes.

“Similarly, in the Cavalli dataset, NFIL3 expression was lowest in the SHH group compared to the other three subgroups (F = 106.98, p = 9.10 x 10-58)”

Response:  The NFIL3 data was not included in the Swartling batch corrected dataset. Hence it was not available.  Therefore in Fig.  I0 have added a panel for NFIL3 data from the original Cavalli dataset and refer to it as Fig.  10C.

Response:

  1. Comment: Please correct the formatting on Fig. 8 legend

Response:  Done

  1. Comment: For NR1D1, the most ….Its Fig 9 instead of Fig 8.

                Response:  corrected

  1. Comment: In the manuscripot, many figures contain subheadings… Please maintain a consistency. 

Response:  For the bar graphs I have checked for consistency. 

Reviewer 3 Report

Comments and Suggestions for Authors

Vriend and Glogowska investigate the role of core clock genes in medulloblastoma and its subgroups. Using information derived from a number of public datasets, the role of clock genes was investigated and translational opportunities identified. Review of available datasets revealed that clock gene expression was found in all subgroups, though with differences in expression based on subgroup. CRY1, USP2, MYCBP2, and TIMELESS were linked to poor survival, and CLOCK was linked to survival protection.

SIMPLE SUMMARY: Provides an adequate summary of the reason for the study and findings. 

ABSTRACT: Adequately summarizes the hypothesis, methods and findings.

INTRODUCTION: The introduction should start with a sentence or two about what clock genes are.  The introduction otherwise provides a background to medulloblastoma and its subtypes and clock gene expression.  

METHODS: Data acquisition and analysis were adequately presented.

RESULTS: Results were presented for with regard to expression of individual core clock genes and, where pertinent, clinical correlate such as survival., The ubiquitin pathway regulation of clock genes was also investigated and presented adequately.  No comments with regard to presentation of results

DISCUSSION: The discussion adequately summarized the pertinence of the findings with regard to clock gene expression, and most pertinent, how some of these would translate to clinical utility.

CONCLUSION: Adequately summarizes the findings and the role in future studies.

REFERENCES: All references are up-to-date and pertinent to the paper

FIGURES AND TABLES: All figures and tables complement or supplement the narrative without duplicating information.

Author Response

Comment:  The introduction should start with a sentence or two about what clock genes are…

Response:  The following sentence was added.

                Clock genes encode transcription factors and proteins that regulate behavioral and physiological rhythms that have a daily pattern (oscillation) of approximately 24 hours.  These function as biological timekeepers of circadian rhythms. 

Round 2

Reviewer 1 Report

Comments and Suggestions for Authors

I think that after the revision, the article is publishable in its current form. No more comments..